

# CARIB12: A Regional Community Earth System Model / Modular Ocean Model 6 Configuration of the Caribbean Sea

Giovanni Seijo-Ellis[1], Donata Giglio[1], Gustavo Marques[2], and Frank Bryan[2]

[1]Department of Atmospheric and Oceanic Sciences, University of Colorado Boulder, Boulder, CO, USA
[2]Climate and Global Dynamics Laboratory, U.S. National Science Foundation National Center for Atmospheric Research, Boulder, CO, USA

**Correspondence:** Giovanni Seijo-Ellis (giovanni.seijo@colorado.edu)

**Abstract.**

A new CESM/MOM6 ocean-only regional 1/12 ° configuration of the Caribbean Sea is presented and validated. The model configuration was developed as a response to the rising need of high-resolution models for climate impact applications. The configuration is validated for the period covering 2000-2020 against ocean reanalysis and a suite of observation-based datasets. Particular emphasis is paid to the configuration's ability to represent the dynamical regime and properties of the region across sub-seasonal, seasonal and inter-annual timescales. Near-surface fields of temperature, salinity and sea-surface height are well represented. In particular the seasonal cycle of sea-surface salinity and the spatial pattern of the low salinity associated with the Amazon and Orinoco river plumes is well captured. Surface speeds compare favorably against reanalysis and show the mean flows within the Caribbean Sea are well represented. We performed a case study to examine the origin of waters arriving to the Virgin Islands Basin and show the model reproduces known pathways and timing for river plume waters intruding the region. The seasonal cycle of the mixed layer depth is also well represented with biases of less than 3 m when comparing to ocean reanalysis. The vertical structure and stratification across the water column is represented favorably against ship-based observations with the largest simulated biases in the near-surface water mass and the sub-surface salinity maximum associated with the sub-tropical underwater mass. The temperature and salinity variability of the vertical structure is well represented in the model solution. We show that mean ocean mass transports across the multiple passages in the eastern Caribbean Sea compare favorably to observation-based estimates, but the model exhibits smaller variability compared to ocean reanalysis transport estimates. Furthermore, a brief comparison against a 1° CESM global ocean configuration shows that the higher resolution regional model improves significant biases in sea-surface salinity and mixed layer depth in particular. Additionally, the regional model better represents important features and variability within the Caribbean Sea compared to the 1° model. Overall, the regional model reproduces to a good degree the processes within the Caribbean Sea and opens the possibility of regional ocean climate studies in support of decision making within CESM.

## 1 Introduction

The Caribbean Sea (CS) is a crucial pathway for ocean circulation. Understanding the dynamics of the Caribbean Sea holds profound implications for global climate patterns (Schmidt et al., 2004; Restrepo et al., 2019; Gradone et al., 2023; Torres





et al., 2023) and regional marine biodiversity (Miloslavich et al., 2010; Bowen et al., 2013; Chollett et al., 2012). The inflow to the eastern CS is dominated by two sources: a) North Atlantic waters, and b) Tropical Atlantic waters modulated by the large freshwater runoff from the Amazon and Orinoco rivers (Johns et al., 2002; Coles et al., 2013; Hormann et al., 2015; Grodsky et al., 2015; Gradone et al., 2023; Seijo-Ellis et al., 2023). The transport of waters into the eastern CS (from here on referred to as inflow) via the passages along the greater and lesser Antilles exhibits variability across different time scales. At the

daily to monthly time scales, the near-surface transport may be driven by surface winds (Johns et al., 2002; Andrade-Amaya, 2000) and/or particular river plume events (Seijo-Ellis et al., 2023). The interactions between topography (along the islands in the eastern CS) and density fronts of plume waters result in baroclinic instabilities that often develop into mesoscale eddies that transverse the CS (Andrade and Barton, 2000; Southwick et al., 2016). At the seasonal scale, the near-surface transport and salinity of the CS are dominated by the runoff from the Amazon and Orinoco river plumes peaking in boreal spring and

reaching its minimum in early fall (Corredor and Morell, 2001; Johns et al., 2002; Jouanno et al., 2012; Seijo-Ellis et al., 2023). At inter-annual to longer timescales, the flow and properties of the waters entering the eastern CS may be modulated by large scale modes of variability such as the El Niño Southern Oscillation and the Atlantic Meridional Mode via coupled processes between the atmosphere, land and ocean (Restrepo et al., 2014; Arias et al., 2015; Restrepo et al., 2019). Below 100-150 meters, the inflow is modulated at inter-annual and longer time scales and is an important component of the Atlantic

Meridional Overturning Circulation (AMOC) return flow (Wilson and Johns, 1997; Johns et al., 1999, 2002). The transport includes water masses such as Sargasso Sea Water, Subtropical Underwater, Antarctic Intermediate Water, South Atlantic Water, and North Atlantic Deep Water (Corredor and Morell, 2001; Johns et al., 2002; Seijo-Ellis et al., 2019; Gradone et al., 2023). Recent studies suggest that the deep transport of South Atlantic Waters through the Anegada Passage may be larger than originally estimated (Gradone et al., 2023). These studies indicate that the inflow of waters to the Caribbean Sea is an important

component towards understanding changes to the AMOC under a changing climate (Bryden et al., 2005; Frajka-Williams et al., 2019; Caesar et al., 2018).

The Amazon and Orinoco river plumes are also responsible for variability within the Caribbean Current via the generation of vertical shear (Chérubin and Richardson, 2007) that accelerates its mean flow. In addition, the strong near-surface stratification due to the plumes limits vertical mixing and can have important implications for the formation and evolution of tropical storms

in the CS by modulating heat fluxes and exchanges with the atmosphere (Godfrey and Lindstrom, 1989; Lukas and Lindstrom, 1991; Rudzin et al., 2017). While the inflow modulates temperature and salinity variability in the CS, there are other important local processes that contribute to upper ocean variability in the basin. For example, tides across the CS drive local vertical mixing and enhance primary productivity (Kjerfve, 1981; Giese et al., 1990; Sosa, 2001; Ezer et al., 2011) in different areas within the CS. Alongshore winds drive upwelling along the southern CS via Ekman transport (Andrade-Amaya, 2000; Digna

et al., 2018) and mesoscale eddies can lead to the formation of thermohaline staircases and mixing via double-diffusion (Morell et al., 2006).

Studying these processes for the present day is a relatively straightforward endeavor combining observations and regional ocean models, but at climate scales additional challenges may arise (Ezer et al., 2011; Solano et al., 2018; Mukherjee et al., 2023; Seijo-Ellis et al., 2023). Coupled climate models have been able to reproduce past and present trends in the Earth System





becoming a powerful tool to examine mechanisms and future changes (Hurrell et al., 2013; Meehl et al., 2014; Kay et al., 2015). Nevertheless, the low spatial resolution does not allow to resolve processes at fine scales, and the dependency on model parameters result in uncertainties across models and against observations (Tebaldi and Knutti, 2007; Hourdin et al., 2017). For example, the typical horizontal resolution for the ocean component in coupled climate models is of 0.25° or coarser, which does not resolve the numerous passages in the eastern CS, nor the processes that are so intrinsically connected to the topography

and geography of the region. Furthermore, certain longstanding biases in climate simulations can often be traced to the ocean component of the coupled model because important oceanic processes are unresolved (or poorly represented) resulting in biases in ocean heat fluxes, air-sea fluxes, and gradients along temperature and density fronts (Large and Danabasoglu, 2006; Kirtman et al., 2012; Danabasoglu et al., 2014; Roberts et al., 2016). More recently, Richards et al. (2021) show that changes in the mixed layer depth and buoyancy fluxes under climate change may result in reduced submesoscale activity which, even in the

presence of appropriate parameter tuning, may be an additional source of biases in the ocean for climate projections. While there is a rising need for regional climate research to inform decision making, the shortcomings of global climate models make them unsuited for these purposes. Thus, dynamically downscaling ocean climate projections using a high-resolution ocean model is an enticing proposition to properly examine regional ocean impacts under a changing climate (Chamberlain et al., 2012; Brickman et al., 2021; Richards et al., 2021).

The Community Earth System Model 2 (CESM, Danabasoglu et al. (2020)) uses the Parallel Ocean Program (POP) 2 as its ocean component. However, the Modular Ocean Model 6 (MOM6, Adcroft et al., 2019) is set to replace POP in future versions of CESM. Meanwhile, MOM6 has been made available to users in release versions 2.2 and higher of CESM2. MOM6 uses innovative approaches that make it an ideal model for high resolution configurations at the regional and global levels. For example, the use of the Arbitrary Lagrangian-Eulerian (ALE) method allows for the use of several different vertical coordinate

systems (geopotential, isopycnal, terrain-following or hybrid), making the model adaptable to the specific constraints of the topography of a region and purposes of the configuration. The model can be run in fully coupled global configurations within CESM, as well as in ocean-only, and other supported combinations. In addition, it provides the user the ability to dynamically downscale the ocean component in global simulations while remaining within the CESM framework. As a first step towards this goal, here we present an ocean only CESM/MOM6 configuration of the Caribbean Sea. This configuration (here on referred to

as CARIB12) was developed to understand present and future variability of the near-surface Caribbean Sea waters. This paper focuses on the configuration of the model and a thorough validation for the present day period (2000-2020) as a benchmark for upcoming future climate projections. In Section 2 we describe the model configuration in detail and, the datasets and methods used to validate the model. The validation and results are shown in Section 3 and conclusions are presented in Section 4.

## 2 Data and Methods

### 2.1 Model Description

The CARIB12 configuration is built on a horizontal Arakawa C-grid (Arakawa and Lamb, 1977) at 1/12 ° resolution (≈ 9 km) resulting in 759 x 457 tracer points. The grid extends from -6°S to 32°N and from -35.5°W to -98.5°W (Figure 1a). The Pacific





Ocean is masked to eliminate an unnecessary open boundary and improve computational efficiency. The topography (Figure 1a) was generated using the Shuttle Radar Topography Mission 15+V2.1 dataset (Tozer et al., 2019) and was smoothed using a Cressman weighted interpolation scheme. Manual edits were made to the land-ocean mask and topography to ensure proper width and depth of the numerous passages between the Caribbean islands, the Amazon river delta, and along the coastline in general. In the vertical, a 65 layer $z^*$ (Adcroft and Campin, 2004) grid is used with a maximum spacing of 2.5 m at the surface and increasing grid spacing at depth using a hyperbolic tangent function (to a maximum spacing of 248.7 m above the maximum depth of 6500 m).

CARIB12 uses a baroclinic time step of 800 s and a thermodynamic time step of 1800 s. The background kinematic viscosity of the interior is set to $1 \times 10^{-6} \frac{m^2}{s}$. CARIB12 uses a Smagorinsky horizontal viscosity with a Smagorinsky constant of 0.02 and a bi-harmonic viscosity set to $5 \times 10^8 \frac{m^4}{s}$. The configuration of the horizontal viscosity was determined after exploring different options and combinations within MOM6, including the use of a Laplacian viscosity and different values for the corresponding constants. The final configuration was largely determined by the mean flows into the CS across the multiple passages between the Caribbean Islands. Vertical mixing is specified via the CvMix library (Griffies et al., 2015) with a background diapycnal diffusivity of $1 \times 10^{-6} \frac{m^2}{s}$. CvMix utilizes the K-Profile Parameterization (KPP) of Large et al. (1994) for the boundary layer parameterization, mixing due to double-diffusion and shear-driven turbulence. The parameterization of Fox-Kemper et al. (2011) is implemented for the re-stratification of the mixed layer by sub-mesoscale eddies with a front length scale of $1500m$. Additional details of the physical configuration and choice of parameterizations in CARIB12 are given in the following subsections and summarized in Table 2.

### 2.1.1 Initial and Open Boundary Conditions

Initial conditions are prescribed from the GLORYS12V1 reanalysis (GLORYS12, Jean-Michel et al., 2021). The GLORYS12 data is on a $1/12°$ resolution grid with 50 vertical levels. The model grid has two open boundaries, one to the east, one to the north (Figure 1a). The open boundary conditions are specified daily using temperature, salinity, sea-surface height (SSH), meridional (v) and zonal (u) velocity components from GLORYS12. Ten tidal constituents are specified at the boundaries. The tidal amplitudes and phases were obtained from the TPXO Global Tidal Model 9v5a (Egbert and Erofeeva, 2002). The barotropic flow at the boundaries is treated with a Flather boundary condition (Flather, 1976). The baroclinic component is specified using the Orlanski (1976) boundary condition. The boundary flows are nudged to exterior velocities at timescales of 0.3 days for inflow and 360 days for the outflow. Nudging layers for temperature, salinity and velocities are applied to minimize noise at the boundaries that may contaminate the interior. The layers are based on mean monthly fields from GLORYS12. Damping in the nudging layers decays across 15 grid points away from the boundaries (dashed black lines in Figure 1a) with maximum damping applied at the boundary grid cells. Damping at each point $i$ is given by: $damping_i = \frac{1}{dmin+dr*i}$, where $dr = (dmax - dmin)/(npts + 1)$, $dmax = 20$ days, $dmin = 2$ days and $npts = 15$ (corresponding to the number of points across which damping decays). The parameters for the applied damping were determined as a balance between reducing noise at the open boundaries and not influencing the solution towards the center of the domain.



### 2.1.2 Additional Forcings

The atmospheric forcing is prescribed from the Japanese 55-year reanalysis (JRA55-do, Tsujino et al., 2018) and is specified every 3 hours. Surface air temperature, downwelling shortwave radiation, downwelling longwave radiation, zonal and meridional wind, specific humidity, sea-level pressure, and precipitation all referenced to 10 m above sea-level in combination with
SST and surface velocities are used to compute surface fluxes of heat, water and momentum using the Large and Pond (1981) bulk formula. Freshwater runoff input is generated using the Global Flood Awareness System (GloFAS, Zsoter et al., 2021). The CESM Climatological Data River model was modified to use with the GloFAS data. The daily runoff is specified every 3-hours with the atmospheric forcing and is spread horizontally across a maximum radius of 600 km radius from the grid point the runoff is prescribed with an e-fold scale of 200 km. A monthly climatology of surface Chlorophyl-a (Chl-a) is provided
from the SeaWIFS satellite mission (NASA Ocean Biology Processing Group, 2018) for the Manizza et al. (2005) opacity scheme.

### 2.2 Validation datasets

For the model validation we use several datasets. These include the same GLORYS12 Ocean reanalysis (Jean-Michel et al., 2021) used for the open boundaries and for the initial conditions. The reanalysis uses the CORAv4.1 database (Cabanes et al.,
2013) to assimilate observations of sea-level anomalies, sea surface temperatures (SST), and *in-situ* profiles of temperature and salinity from Argo floats (Argo, 2000), shipboard XBT and CTD. Daily fields of temperature, salinity, mixed layer depth (MLD), sea-surface height (SSH) and velocities are used to validate CARIB12. Additionally we validate the mixed layer depth (calculated using the $\Delta 0.03 \frac{kg}{m^3}$ density criterion) with the climatology of deBoyer Montégut (2023). Altimetry derived geostrophic currents from the Copernicus Marine Service (Mercator Ocean, 2021) are used to validate the surface velocities
and eddy kinetic energy field. The eddy kinetic energy was calculated as follows: $EKE = (\frac{1}{2})(u'^2 + v'^2)$, where $u'$ and $v'$ are velocity anomalies resulting from decomposing the velocities into a mean velocity ($\overline{u}$ and $\overline{v}$) and the anomaly: $u = \overline{u} + u'$. Further validation is done for sea-surface temperature (SST) fields against the Optimum Interpolation SST dataset (Huang et al., 2021) and, for sea-surface salinity (SSS) fields, against the Multi Observation Global Ocean Sea Surface Salinity product (Droghei et al., 2016). The SST, SSS and MLD are also compared to the 1 ° resolution CESM-POP configuration used for the
Ocean Model Intercomparison Project phase 2 (Tsujino et al., 2020). We have additionally validated water mass representation in the model against ship-based observations (CCHDO Hydrographic Data Office, 2023), using the Argovis application programming interface (Tucker et al., 2020) to identify WOCE lines within the region of interest (i.e. WOCE lines A20 and A22, Figure 1b) and specific cruises within the time frame of the simulation (2000-2020). Profiles of temperature and salinity corresponding to the closest profile in time and space to each ship-based profile were extracted from the CARIB12 solution
and the GLORYS12 reanalysis. The cruise profiles, CARIB12 and GLORYS12 profiles were binned in 0.2 PSU × 1°C bins after being vertically interpolated onto a common grid. When comparing our model to different data products, the dataset with higher resolution was bi-linearly interpolated onto the horizontal grid of the product with lower resolution using the xESMF





Python Package (Jiawei Zhuang et al., 2023). The CARIB12 solution was also saved at run-time on the GLORYS12 vertical grid and the ship-based profiles were re-gridded onto that same vertical grid.

## 2.3 Lagrangian Particle Tracking Model

We perform a case study validation of CARIB12 using the Lagrangian particle tracking model Ocean Parcels (Lange and Sebille, 2017; Delandmeter and Sebille, 2019). We replicate the methodology and experiment performed by Seijo-Ellis et al. (2023) to examine the origin and pathways of waters arriving to the Virgin Islands Basin (VIB). One hundred passive particles were released daily at the grid point closest to $18°N$, $64.8°W$; i.e. the closest point to site Bi14 in ][]SEIJO2023 and backtracked for 100 days. Backtracking is done via a 4th order Runge-Kutta interpolation scheme using the CARIB12 daily surface velocity fields. This approach allows to trace the trajectories of the particles 100 days before their arrival to the release site. We focus on years 2009, 2010 and 2011 (only 2010 is shown), in order to compare to the results shown in Figures 5 and 6 of Seijo-Ellis et al. (2023).

The analysis of the trajectories follows the same method described in Seijo-Ellis et al. (2023). For each arrival date (same as release date when backtracking), we count the number of trajectories within $0.2° \times 0.2°$ bins. Two main constraints are enforced: particles with more than one position recorded in the same bin are only counted once, if a particle passes through but has no recorded position in a particular bin it is not counted. A probability density map of the trajectories of the particles 100 days prior to their release is then generated for each release date (Figure A1a and b). In order to summarize the temporal evolution of these binned trajectory maps, we generate a time series of the cumulative number of trajectories per bin color-coded by each bin's location (Figure A1 and 16). The first step is to color code the bins by their location such that a change in color represents a meridional change in position, and a change in color shading represents a latitudinal change in position (Figure A1c). For each arrival date, color-coded bins are stacked on top of each other according to the number of particles in each bin and scaled according to the number of bins with particles (Figure A1d). The resulting time-series shows the pathways of waters arriving to the release site (Virgin Islands Basin) on any given date (Figure A1e). For a detailed description of the method please refer to Seijo-Ellis et al. (2023).

## 3 Results

### 3.1 Near-surface Fields

In the following subsections we examine temperature, salinity, sea level and kinetic energy in the shallowest layer of the model (0-2.5 meters) within the domain of interest (the region shown in 1b), as described in Section 2. Our focus is on time averaged fields, with the average computed for the full time series or specific seasons.





### 3.1.1 Temperature and salinity

Within the domain of interest (Figure 1b), CARIB12 has a mean surface temperature of 27.5 °C with a spatial standard deviation of 0.52 °C (Figure 2a). The spatial patterns of temperature across the domain are well represented and CARIB12 captures the upwelling system in the southern CS. The GLORYS12 reanalysis mean SST is 27.6°C with a standard deviation of 0.56°C (Figure 2b). These statistics indicate good agreement and similar distributions between CARIB12 and the GLORYS12 reanalysis, with a small cold bias of -0.15°C (Figure 2c). The OISST data has a mean SST of 27.6°C and standard deviation of 0.55°C (Figure 2d). CARIB12 shows biases against OISST similar to those in the comparison to GLORYS12, the mean bias is -0.15°C being dominated by biases in shallow areas and within the CS (Figure 2e).

CARIB12 compares favorably against both datasets not only in terms of time mean, but also for seasonal changes: correlation very close to one is found between the CARIB12 12-month climatology and the 12-month climatology based on other products (Figure A2); also, differences between the CARIB12 standard deviation for the 12-month climatology and the standard deviation in other products are generally within $\approx 10\%$ of the values in the products used for validation (Figure A3). The low resolution CESM-POP simulation represents the broad patterns and mean SST within the region of interest (Table 3). However, it lacks proper representation of the extent and magnitude of features characteristic of the CS such as for example, the magnitude of the upwelling system off the coast of South America (Figure A4a).

The spatial patterns of SSS in different seasons are consistent between CARIB12 and both GLORYS12 and the MOGO observational dataset (Figures 3 and 4). Figure 3 shows the mean winter (DJF) SSS for CARIB12 and the validation datasets: the spatial mean in CARIB12 is 35.86 PSU with a mean bias of 0.14 PSU compared to GLORYS12 (Figures 3a and c), and a bias of 0.05 PSU compared to the MOGO observational dataset (Figure 3e). Notably, GLORYS12 appears to have a freshwater source in the region of the Dominican Republic and Puerto Rico resulting in positive biases within the CS. This is highlighted by the red contour line corresponding to 35 PSU, shown in Figure 3b. During the summer (JJA), CARIB12 shows a mean bias of 0.17 PSU compared to GLORYS12 and one of -0.12 PSU compared to MOGO. The 35 PSU contour line (red line in Figures 3 and 4) delineates the fresh water salinity front (based on the discussion in Seijo-Ellis et al., 2023) and shows that the spread and extent of the plume waters is similar between CARIB12 and GLORYS12, but is much smaller in the gridded observations (particularly during the winter, Figure 3).

The seasonal spread of the low salinity plume from the Amazon and Orinoco rivers is well captured with peak high salinity during the winter months within the CS and minimum salinities during the summer months. The largest biases in SSS are the areas of major riverine inputs: the Amazon and Orinoco rivers. These biases may arise from the specification of the river runoff to the ocean and how mixing of the freshwater input in the ocean is handled in CARIB12 (and e.g. differences with the corresponding choices in GLORYS12). In CARIB12, runoff is not distributed vertically in the ocean but rather spread horizontally across a maximum radius of 600 km with an e-fold decay scale of 200 km at the shallowest layer. Also, some of the large biases when comparing to observations may arise due to limited observations in the shallow shelf regions and the interpolations done to grid the data. Nevertheless, the seasonal SSS is well represented in CARIB12 with good correlations between datasets and smaller biases within the Caribbean Sea away from riverine sources. The smallest biases are generally





found within the CS which indicate CARIB12 is performing well within the region of interest. Correlation close to one is found
between the CARIB12 12-month climatology and the 12-month climatology based on other products, in most of the domain
(Figure A5); also, within the CS, differences between the CARIB12 standard deviation for the 12-month climatology and the
standard deviation in other products are generally within $\approx 25\%$ of the values in the products used for validation, except in the
western part of the basin (Figure A6).

In contrast to the well simulated variability of SSS in CARIB12, the low resolution CESM-POP simulation exhibits biases
that hinder the model's performance to properly represent the SSS variability within the Caribbean Sea (Table 3 and Figure
A4b and c). There is an overall positive salinity bias irrespective of the season and the extent and magnitude of the Amazon
and Orinoco river plumes are not well represented. These biases may be a product of several factors: biases in the specification
of the river runoff, inadequate representation of sub-grid scale process and mesoscale eddies that play an important role in the
230 spreading of the river runoff and salinity variability in the region, and/or biases inherent to the low resolution of the model.

### 3.1.2 Surface currents speed, eddy kinetic energy and sea surface height

The mean surface speed is shown in Figure 5. The mean speed in CARIB12 is 15.6 cm/s which corresponds to a mean bias of
$\approx 1$ cm/s below the mean speed in GLORYS12. The largest simulated biases correspond to the region southwest of Hispaniola
where two jets converge: the main Caribbean Current and the jet around $18°$N (Chérubin and Richardson, 2007). The biases in
this area are in the order of up to 20 cm/s. This region is one of increased shear with rapidly varying bottom topography. Thus,
difference in the topography between the models and the specification of the horizontal viscosity may be driving these biases.
The speed compares well elsewhere in the region of interest including the narrow passages in the eastern CS. This indicates
that CARIB12 represents well the inflow to the CS and the flows associated with the advection of the freshwater plumes from
the Amazon and Orinoco rivers. The biases are larger when comparing CARIB12 to the altimetry-based GOGSSH (mean
bias of 3.8 cm/s), but it is worth noting the difference in spatial resolution and that the altimetry derived velocities are purely
geostrophic which may not be a good approximation in this region (surface Ekman currents can reach 50 cm/s in parts of the
Caribbean Basin, Andrade-Amaya, 2000). The seasonal variability of the flow speed is well represented in CARIB12 with
higher mean speeds in the summer and the lowest mean speeds during the winter (not shown), and this is evidenced by the eddy
kinetic energy fields (Figures 6 and 7). In particular, CARIB12 captures the strengthening of the Caribbean current during the
245 boreal summer months (JJA) as well as the re-circulation that occurs within the Colombian basin.

    The seasonal eddy kinetic energy (EKE) for CARIB12, GLORYS12 and from altimetry derived geostrophic currents is
shown in Figures 6 and 7: an overall mean negative bias is observed irrespective of the season and is most notable in the
summer, when EKE is larger (Jouanno et al., 2012). These biases are particularly noticeable south of Hispaniola and in the
southwest corner of the CS where a re-circulation of the surface flow occurs. Despite the lower EKE, seasonal variability is
250 well represented in CARIB12 and the spatial variability compares well with GLORYS12 and the altimetry derived product
(Figures 6c-e and 7c-e). The seasonal increase in EKE during summer months is captured, although mesoscale patterns are not
well represented. In particular, the EKE hotspot extending from the eastern Caribbean Sea through the Colombian Basin is not
as strong in CARIB12 as GLORYS12 and altimetry suggest. The 20 year time-mean EKE biases (not shown) are of the same





order of magnitude as the seasonal EKE biases which indicate that the seasonal biases are dominated by biases in the long term mean. Further improvements in parameter choices may help reduce some of these biases and improve representation of mesoscale variability.

Figure 8 shows the winter and summer mean SSH for CARIB12 and GLORYS12. The winter mean SSH in CARIB12 is 0.12 m and compares well with the 0.17 m mean in GLORYS12 (Figure 8c-f); similarly, a good comparison is seen for the summer (Figure 8c-f). The SSH within the region of interest is characterized by a negative SSH structure along the shelf waters of South America which delineates the Caribbean Current. The meridional extent of this feature extends further off-shore in CARIB12 indicating a wider Caribbean Current in CARIB12 compared to GLORYS12. GLORYS12 has consistently larger mean SSH overall but spatial variability is consistent between the two, as shown by the standard deviations and correlations (Figure 8). CARIB12 also reproduces well the seasonality of SSH, with higher mean SSH and variability during the boreal summer compared to winter. This seasonal cycle is intrinsically connected to the seasonal variability of the mean speed and SSS (both driven by the freshwater plumes of the Amazon and Orinoco rivers).

## 3.2 Mixed layer depth

We compare the winter and summer mixed layer depth (MLD) against the GLORYS12 reanalysis and the climatology by deBoyer Montégut (2023). As detailed in Section 2, the calculation is done using the $\Delta 0.03 kg/m^3$ density criterion with respect to surface values in CARIB12 and GLORYS12, and with respect to a depth of 10 m in the deBoyer Montégut (2023) climatology. CARIB12 compares favorably against GLORYS12 (Figures 9 and 10). The mean bias in the winter is 2.58 m, indicating an overall deeper MLD in CARIB12, with a RMSE of 6.43 and correlation of 0.88. In the summer, CARIB12 has a slightly shallower MLD than GLORYS12, with a mean bias of -0.14 m, RMSE of 3.71 and correlation of 0.89. The biases between CARIB12 and GLORYS12 are attributable to the small biases described for salinity: an overall positive salinity bias (Figures 3 and 4) corresponds to saltier waters in the near surface which leads to weaker vertical stratification (Figure 12) in the upper 0-100 m resulting in a deeper mixed layer particularly during the winter.

When comparing the MLD against the deBoyer Montégut (2023) climatology, CARIB12 has an overall shallower MLD across seasons. During the winter, the mean MLD bias is -5.24 m, with a RMSE of 9.53 and correlation of 0.78. During summer, the bias is -6.58 m, with a RMSE of 8.03 and correlation of 0.83. CARIB12 represents well the MLD seasonality across the domain: the mixed layer is deepest in winter and becomes shallower during the spring as temperatures rise and the Amazon and Orinoco river plumes influence the Caribbean region. The mixed layer is overall shallowest during the summer months before deepening again in the fall as near-surface temperatures decrease and salinity increases in the absence of the Amazon and Orinoco river plumes. Overall, CARIB12 captures both the seasonal variability and spatial patterns of the MLD and compares well to GLORYS12. While CARIB12 also compares favorably to the deBoyer Montégut (2023) climatology, one must exercise caution in this comparison as the CARIB12 MLD is calculated using the $\Delta 0.03 kg/m^3$ criterion referenced to the shallowest layer in the model (1.25 m), while the deBoyer Montégut (2023) dataset is calculated referenced to 10 m depth.





The CESM-POP simulated MLD shows biases in the order of 13 m during the summer and 30 m during the winter for the mean MLD (Table 3). These biases in CESM-POP could be in part attributed to biases in salinity (Section 3.1.1): a positive salinity bias leads to weaker vertical stratification and thus an overall deeper mixed layer. While the broad spatial patterns of MLD during the summer agree with those in CARIB12 and other datasets, the patterns during the winter do not appear to be as well represented in the CESM-POP simulations and further highlight the important role of Amazon river runoff and salinity in the Caribbean Sea (Figure A4d and e).

### 3.3 Vertical structure and water mass properties

We compare the vertical structure of the water column in CARIB12, with GLORYS12 and with available observations along WOCE lines A20 and A22 within the region of interest (Figure 1b). Figure 11 shows 2D histograms of temperature and salinity for CARIB12, GLORYS12 and GO-SHIP CTD data (Figure 11a-c, respectively; CARIB12 and GLORYS12 data are co-located in space and time with profiles along the WOCE lines): the overall vertical structure and water mass characteristics is well represented in CARIB12, with some of the largest differences occurring in the shallower Caribbean surface waters (Figure 11d-e). These differences are not surprising, surface waters exhibit higher variability as they are exposed to a number of large scale circulation and forcing mechanisms, e.g. the influence of river runoff at sub-seasonal to seasonal scales, wind-driven circulation, eddies and the Caribbean Current (Corredor and Morell, 2001; Jouanno et al., 2012; Digna et al., 2018; Seijo-Ellis et al., 2023). Additional biases are noted around the salinity maximum (Subtropical Underwaters, as described in e.g. Seijo-Ellis et al., 2019; Torres et al., 2023) where there is a wider spread in values in CARIB12 compared to GLORYS12 and GO-SHIP. Nevertheless, the biases in the sub-surface salinity maximum are small and the model shows good agreement with GLORYS12 and observations. These differences may reflect larger variability in the sub-surface inflow associated with the North Atlantic gyre and/or the formation of the SUW mass in the North Atlantic.

A comparison between CARIB12 and GLORYS12 for the area-weighted time mean profile of salinity and temperature in the region of interest (Figure 12) shows a salty and slight cold bias near the surface (consistent with the discussion in Section 3.1). At depth, the biases are reversed with a fresh bias in the 100-150 m depth range, which corresponds to the sub-surface salinity maximum associated with Subtropical Underwaters, and a warm bias of up to 0.5°C. Biases in the mean profiles of temperature and salinity get smaller with depth (Figure 12). Biases seen in Figure 12 are representative of the biases in Figure A7, which shows that CARIB12 reproduces well the large scale signals in both temperature and salinity across the length of the simulation.

The area weighted seasonal cycle of salinity and temperature in CARIB12 is in good agreement with GLORYS12 (Figure 13). Model biases indicate the amplitude of the seasonal cycle in CARIB12 is well represented compared to GLORYS12. The near-surface biases in the seasonal cycle of temperature and salinity are of the same order as those shown in Section 3.1. Figure 13 shows the strong seasonal cycle in salinity for the region that is largely driven by the runoff from the Amazon and Orinoco rivers during late boreal spring through early fall and the higher salinity during late fall through winter into early spring driven by the lower riverine input to the ocean and the inflow of saltier Atlantic waters. These results highlight CARIB12's ability to correctly represent temperature and salinity variability in the CS as seasonal scales. At inter-annual timescales, CARIB12





salinity and temperature anomalies in the upper 300 m are overall consistent with GLORYS12 (A7). The largest anomalies in GLORYS12 are present in CARIB12, including signals likely associated with the 2015/2016 La Niña (salty and warm anomalies from 2015 onwards, Jiménez-Muñoz et al., 2016).

## 3.4 Transports

Ocean mass transports were calculated for passages in the eastern and northern Caribbean Sea. The passages are shown in Figure 1b and a summary of the mean transports in CARIB12, GLORYS12 and observational estimates can be found in Table 4 (for each of the passages of interest). We note that while transports based on CARIB12 and GLORYS12 are a time mean during 2000-2020, observational estimates are available only for shorter periods of time.

The mean transport across the Windward Passage between Cuba and Hispaniola is 1.91 Sv into the Caribbean Sea which is
330 lower than that in GLORYS12 (2.8 Sv) and observational based estimates (3.8/3.6 Sv) by Smith et al. (2007). Nevertheless, Johns et al. (2002) and Smith et al. (2007) indicate there is still uncertainties on the transport through this passage.

The inflow across the Mona Passage (0.38 Sv) is also underestimated in CARIB12 compared to observations (3.0 Sv); here, GLORYS12 has a net outflow of 1.22 Sv which, along with the freshwater shown for GLORYS in Figure 3b (red contour in the region of the Domincan Republic and Puerto Rico), may be a significant contributor to the biases seen in the region when
comparing CARIB12 to GLORYS12 (e.g. Figure 3c).

Flow through the Anegada Passage is an important contributor to the circulation between the Atlantic and the Caribbean Sea: CARIB12 shows good agreement with observations by Johns et al. (2002) with a 1.53 Sv inflow compared to 2.5 Sv. However, recent studies suggest that the inflow through the Anegada Passage may be larger than previously thought (e.g., Gradone et al., 2023). CARIB12 captures the mean transport through the Anegada Passage better than GLORYS12 which has a mean outflow
of 0.14 Sv.

The Antigua Passage mean inflow is 2.11 Sv in CARIB12 which compares favorably against 3.1 Sv estimated by Johns et al. (2002) and is about half the transport in GLORYS12 (-4.06 Sv). Across the Guadeloupe Passage the inflow in CARIB12 (0.74 Sv) agrees better with observations (1.1 Sv) than GLORYS12 (0.10 Sv). Similarly for the Dominica Passage, although both CARIB12 (2.79 Sv) and GLORYS12 (3.02 Sv) overestimate the inflow here compared to 1.6 Sv inflow estimated by Johns
et al. (2002). The St. Lucia Passage mean inflow in CARIB12 agrees well with observations (1.97 Sv in CARIB12 and 1.5 Sv from observations) while GLORYS12 overestimates the inflow by 3.4 Sv.

The St. Lucia - Trinidad section (Figure **??**) is a combination of the St. Vincent and Grenada Passages in Johns et al. (2002). CARIB12 has a mean inflow of 9.51 Sv which agrees well with the 8.6 Sv estimate by Johns et al. (2002). Along this section, GLORYS12 has a mean inflow almost twice as large (16.62 Sv) as suggested by observations, which may also contribute to
350 near-surface biases in the CS (particularly the southern part) discussed in Sections 3.1 and 3.1.2.

The net mean inflow to the Caribbean Sea is 20.94 Sv which is 0.31 Sv more than the mean flow out of the Caribbean Sea via the Yucatan Channel (20.63 Sv) (Figure 1a). The mean flow through the Yucatan Channel in CARIB12 is slightly less than that estimated from observations by Sheinbaum et al. (2002) and Candela et al. (2003) (23.8 and 23.06 SV, repectively). It is worth noting that there are open passages between the Anegada and Antigua Passages, as well as the strait between South



America and the island of Trinidad and Tobago which may result in higher net inflow. In addition, the section defining the
      Yucatan Channel in CARIB12 is not completely bounded by land which may result in a lowe mean outflow there.

      At sub-seasonal timescales, CARIB12 shows less variability compared to GLORYS12 (Figure 14), which is reminiscent of
      the negative biases in surface speed and EKE described in Section 3.1.2. At seasonal timescales, there is an overall agreement
      between CARIB12 and GLORYS12 (Figure 15a-f). With the exception of the Guadeloupe and Dominica Passages (Figure
15c and d), GLORYS12 shows stronger seasonal variability than CARIB12. We note that, as the seasonal cycle is regulated
      by the seasonal freshwater runoff from the Amazon and Orinoco river plumes (Wilson and Johns, 1997; Corredor and Morell,
      2001; Johns et al., 2002; Chérubin and Richardson, 2007; Jouanno et al., 2012; Seijo-Ellis et al., 2023), differences in how the
      runoff is represented in the two models (runoff is applied in the subsurface in GLORYS12 vs at the surface in CARIB12) may
      have implications for the across-passage transports and thus result in differences between CARIB12 and GLORYS12. Finally,
GLORYS12 shows stronger inter-annual variability than CARIB12 (Figure 15g-l). While the mean inter-annual flows are well
      represented in CARIB12, the model does not capture the same amplitude of variability that GLORYS12 suggests exists in
      some the passages (Figures 15g-l). These differences may be further indicators of the differences between the models: runoff
      specification, horizontal viscosity configuration and the overall differences in model forcing (i.e. GloFAS runoff in CARIB12
      versus runoff climatology in GLORYS12). Continuous observations would be needed to better assess how CARIB12 represents
variability across timescales in the transports across the numerous passages. Nevertheless, both CARIB12 and GLORYS12
      indicate that the flow across these passages is complex and highly variable which has been highlighted by previous studies and
      remains an active area of research (Wilson and Johns, 1997; Johns et al., 1999; Centurioni and Niiler, 2003; Johns et al., 2002;
      Gradone et al., 2023).

### 3.5   Intrusions of the Amazon River Plume in the Caribbean and the salt budget

Salinity variability in the Virgin Islands Basin (VIB) is consistent with that of the wider Caribbean Sea and shows a strong
      seasonal component: salinity remains above 35.5 PSU during the first half of the year and below 35.5 PSU during the second
      half (Figure 16a, red line), consistent with the results for the wider CS in Sections 3.1 and 3.1.1. To test CARIB12's ability
      to replicate known pathways of low salinity waters in the Caribbean Sea (modulating the seasonal variability of salinity in the
      region), we designed an experiment similar to that in Seijo-Ellis et al. (2023).

Figure 16a shows that the seasonal decrease in near-surface salinity is largely driven by near-surface horizontal salinity
      advection ($-u\frac{\partial S}{\partial x} - v\frac{\partial S}{\partial y}$, blue line in the figure) as that contributes to much of the change in the near-surface salinity tendency
      ($\frac{\partial S}{\partial t}$, black line). Variability in salinity advection is associated with intrusions of Amazon river waters: salinity starts decreasing
      between May and June, as Amazon river plume waters arrive into the VIB (as indicated by the light blue colors in Figure 16c).
      The lack of light blue and green for most of the first half of the year indicate that waters arriving at the VIB were coming
from the dark black, purple and red regions in Figure 16b. That agrees with previous studies showing that during winter and
      early spring, waters arriving at the VIB are saltier and slower moving than those during the second half of the year when
      near-surface horizontal advection is stronger (Figure 16a). During the second half of the year, when advection is stronger
      and more variable (e.g. September - October, 2010) waters arriving at the VIB followed a path from the Amazon river delta



(light blue) northwestward along the shelf of South America (light green, orange and red) into the Caribbean Sea (light to
390 medium purple) and to the release site in the VIB (blacks). Results here are consistent with findings by Seijo-Ellis et al. (2023)
regarding the seasonality of salinity and the arrival of plume events in the VIB, and how near-surface horizontal advection plays
a dominant role in the salt budget of the VIB. These results highlight the control changes to freshwater runoff from the Amazon
and Orinoco river may have on the salinity of the Caribbean Sea as a dominant process for salinity variability. In particular,
these results show that CARIB12 reproduces known patterns of near-surface horizontal advection which are important to study
additional processes in the CS.

## 4  Conclusions

A new regional ocean-only CESM/MOM6 configuration of the Caribbean Sea for the present day (2000-2020) is validated
against the GLORYS12 reanalysis and a suite of observation-based datasets. Near-surface fields of temperature, salinity and
SSH are well represented, with small mean biases, comparable standard deviations and high correlations between CARIB12,
GLORYS12 and the respective observation based datasets for each field. In particular, the seasonal cycle of SSS and the spatial
pattern of the low salinity associated with the Amazon river plume is well captured. Eddy Kinetic energy shows some negative
biases within the Caribbean Sea, yet the seasonal variability is captured. The MLD is also well represented, with a deep winter
mixed layer and a shallow summer MLD, and biases against the GLORYS12 reanalysis generally less than 3m. Further tuning
of the horizontal viscosity and choice of parameters may be needed to improve the biases in near-surface fields, particularly
mesoscale variability across the Caribbean Sea.

The vertical structure of the water column is well captured in CARIB12 compared to GLORYS12 and GO-SHIP data.
The largest differences are observed in the near-surface water mass and in the layer of the sub-surface salinity maximum
which, in CARIB12, is broader and not as pronounced as in the GLORYS12 reanalysis and GO-SHIP data. Nevertheless, the
overall vertical structure is reproduced accurately in CARIB12. Vertical temperature and salinity variability is also overall
well represented across timescales. While some differences exist at sub-seasonal timescales, the seasonal cycle is represented
well. Across the full simulation, anomalies in temperature and salinity across the water column are comparable to those in the
reanalysis.

Ocean mass transports across a number of passages in the northern and eastern Caribbean sea are computed and compared
against GLORYS12 and observations. The mean flows in CARIB12 compare favorably against observations while GLORYS12
exhibits some biases, particularly along the Mona Passage, Anegada Passage and the St.Lucia - Trinidad section. The seasonal
and inter-annual transports in CARIB12 compare overall well with GLORYS12, yet GLORYS12 exhibits larger variability
in some cases. GLORYS12 suggests larger variability at sub-seasonal timescales too. Nevertheless, there are no observations
available long enough to assess the variability of the flows along the passages across different timescales.

We performed a case study replicating the analysis by Seijo-Ellis et al. (2023) and find that the pathways and timing of
420 plume intrusions into the Virgin Islands Basin agree well with those in GLORYS12. Furthermore, we show that near-surface
horizontal salinity advection generally drives the near-surface salinity tendency in the Virgin Islands Basin, consistent with



Seijo-Ellis et al. (2023). These results highlight the important role the advection of freshwater runoff from the Amazon and Orinoco rivers has in modulating the near-surface salinity of the Caribbean Sea (consistent with previous studies, e.g. Corredor and Morell, 2001; Johns et al., 2002; Jouanno et al., 2012; Grodsky et al., 2015; Gouveia et al., 2019; Seijo-Ellis et al., 2023) and the ability of CESM/MOM6 to properly represent the processes within the CS.

We compared mean fields for SST, SSS and MLD of CARIB12 against a 1 ° resolution CESM-POP configuration. While the CESM-POP captures the mean state and broad patterns of SST in the CS, it does not represent well significant dynamical features that drive variability in the region. For example, the location, spatial extent and magnitude of both the Caribbean Current and the upwelling system in South America are not well represented. In addition, there are biases in SSS and the representation of the Amazon and Orinoco river plumes. The extent and seasonal amplitude of the SSS cycle is not well captured including unresolved mesoscale and sub-mesoscale instabilities that drive much of the variability within the CS. These salinity biases may also be responsible for large biases seen in the mixed layer depth suggesting that the low resolution does not represent the regional vertical stratification and variability to a good degree. CARIB12 improves these biases significantly, highlighting the need of high-resolution regional models in order to properly address climate impacts at these scales and support decision making.

With this thorough validation we showcase the new capabilities within CESM to perform high resolution regional ocean modeling. CARIB12 and the new regional ocean capabilities of CESM are invaluable tools for regional climate impact studies. As a community model configuration available to the wider scientific community, CARIB12 is a valuable tool for actionable science at regional scales and provides the foundations for further development of regional ocean climate modeling within CESM. For example, as part of a separate project, the CARIB12 configuration will be used for dynamical downscaling of CESM climate projections to examine changes in the heat and salinity budgets of the Caribbean Sea under future climates.

*Code and data availability.* CARIB12 was developed on CESM and model components version $cesm2\_3\_alpha16b$. This and other versions of CESM are published and available via NSF-NCAR at www.cesm.ucar.edu:/models/cesm2/). A copy of the specific version used for CARIB12 is also hosted here: https://doi.org/10.5281/zenodo.11289425. CARIB12 configuration and input files can be accessed here: https://doi.org/10.5281/zenodo.11165669. CARIB12 output fields presented in this manuscript can be accessed here: https://doi.org/10.5281/zenodo.11264010. Trajectories generated with Ocean Parcels v2.0 and CARIB12 surface velocities, as well as the Ocean Parcels code used to generate the trajectories can be found here: https://doi.org/10.5281/zenodo.11267616. The datasets used for model forcing and validation are listed as follows: GLORYS12 reanalysis (https://doi.org/10.48670/moi00021, Global Ocean Physics Reanalysis, 2021); OISST v2 (https://psl.noaa.gov/data/gridded/data.noaa.oisst.v2.highres.html, Huang et al. 2021); mixed layer depth (https://doi.org/10.17882/91774, deBoyer Montégut, 2023); global ocean gridded sea surface heights (https://doi.org/10.48670/moi-00148, Global Ocean Gridded L4 Sea Surface Heights And Derived Variables Reprocessed 1993 Ongoing, 2023); TPXO9 (https://www.tpxo.net/home, Egbert and Erofeeva, 2002); Multi Observation Global Ocean Sea Surface Salinity (https://doi.org/10.48670/moi-00052, Droghei et al. 2016); CCHDO Hydrographic Data (CCHDO Hydrographic Data Office, 2023).



*Author contributions.* Conceptualization: G.S-E., D.G., G.M., F.B., Model configuration: G.S-E., D.G., G.M., F.B., Model simulations: G.S-E, G.M., Model Evaluation: G.S-E., D.G., G.M., F.B., Formal Analysis: G.S-E., D.G., G.M., Visualization: G.S-E., D.G., G.M., Original Draft: G.S-E., Review and Editing: G.S-E., D.G., G.M., F.B.

*Competing interests.* The authors declare no competing interests.

*Acknowledgements.* G.S-E. and D.G. acknowledge support by NSF award 2026954 and NOAA award NA21OAR4310261. Part of the work by G.S-E. was completed under the NSF NCAR ASP-Graduate Visitors Program. This material is based upon work supported by the National Center for Atmospheric Research, which is a major facility sponsored by the National Science Foundation under Cooperative Agreement No. 1852977. We would like to acknowledge high-performance computing support from Cheyenne (doi:10.5065/D6RX99HX) provided by NSF NCAR's Computational and Information Systems Laboratory, sponsored by the National Science Foundation. We would also like to acknowledge valuable input provided by the informal MOM6 Regional Modeling meetings led by Dr. Charles Stock (NOAA/GFDL) and Dr. Andrew Ross (NOAA/GFDL). We recognize valuable input and support from the Earth System Modeling Group at Rutgers University, in particular Dr. Enrique Curchitser, Dr. James Simkins, Dr. Nicole Laureanti, Mr. Rob Cermak, and Mr. Raphael Dussain (Rutgers/GFDL). In addition, we would like to acknowledge valuable assistance and input from Dr. Michael Levy and other members of the Ocean Section at NSF NCAR.



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



**Table 1.** Physical configuration, parameterization and parameter choices for CARIB12.

| Parameter | Value | Reference |
|---|---|---|
| **Time-stepping** | Baroclinic 900s | |
| | Thermodynamic 1800s | |
| **Grid** | | |
| Horizontal | 1/12° | |
| Vertical | 65 layer z* | |
| **Open Boundary Conditions** | | |
| Barotropic | Flather | Flather (1976) |
| Baroclinic | Orlanski (nudging timescales: 0.3 day for inflow and 360 days for outflow) | Orlanski (1976) |
| Tracers | Reservoir length scales: $3 \times 10^4$ m (out), 3,000 m (in) | |
| **Tides** | | |
| Explicit from TPXO | 10 tidal constituents: M2, S2, N2, K2, K1, O1,P1, Q1, MM, MF | Egbert and Erofeeva (2002) |
| **Background kinematic viscosity** | $1.0 \times 10^{-6} \frac{m^2}{s}$ | |
| **Coriolis Discretization** | SADOURNY75 Energy | Sadourny (1975) |
| **Horizontal Mixing** | | |
| Horizontal viscosity | Laplacian | |
| Biharmonic horizontal viscosity | background: $5 \times 10^8 \frac{m^4}{s}$ Velocity ($\frac{m}{s}$) and Time ($s$) scales = 0 | |
| Nonlinear eddy viscosity | Smagorinsky (constant = 0.02) | Griffies and Hallberg (2000) |
| **Vertical Mixing** | CvMix Library | Griffies et al. (2015) |
| Boundary layer | K-Profile Parameterization (KPP) | Large et al. (1994) |
| Background diapycnal diffusivity | $1.0 \times 10^{-6} \frac{m^2}{s}$ | |
| Shear-driven turbulence | LMD94 | Large et al. (1994) |
| Mixing due to double diffusion processes | | Large et al. (1994), Danabasoglu et al. (2006) |
| Prandtl Number | 1.0 | |
| **Nudging Layers** | T, S, u and v from GLORYS monthly means decay time scale minimum = 2 days decay time scale maximum = 20 days | |
| **Mixed Layer re-stratification** | Front length scale = 1500 m Decay time scale = $2.59 \times 10^6$ s | Fox-Kemper et al. (2011) |



**Table 2.** Summary of datasets used in the validation of CARIB12 with their corresponding information. Datasets with a $^*$ next to their name are also used for the model forcing.

| Dataset | Fields | Reference | Resolutions |
|:---:|:---:|:---:|:---:|
| GLORYS12v1 (GLORYS12)$^*$ | temperature, salinity, velocities, sea-surface height mixed layer depth | Jean-Michel et al. (2021) | Space: $\frac{1}{12}^\circ$ <br> Time: daily, monthly <br> Depth of first layer = 0.5 m |
| Optimally Interpolated SST v2 (OISST) | sea-surface temperatures | Huang et al. (2021) | Space: $\frac{1}{4}^\circ$ <br> Time: monthly |
| Multi Observation Global Ocean Sea Surface Salinity (MOGO) | sea-surface salinity | Droghei et al. (2016) | Space: $\frac{1}{8}^\circ$ <br> Time: monthly |
| Global Ocean Gridded L4 Sea Surface Heights (GOGSSH) | sea-surface height, geostrophic velocities | Mercator Ocean (2021) | Space: $\frac{1}{4}^\circ$ <br> Time: monthly |
| Deboyer MLD Climatology | mixed layer depth | deBoyer Montégut (2023) | Space: $2^\circ \times 2^\circ$ |
| CESM-POP | sea-surface temperature, sea-surface salinity, mixed layer depth | Tsujino et al. (2020) | Space: $1^\circ$ <br> Time: monthly <br> Depth of first layer = 10m |





**Table 3.** Summary of mean and standard deviation of SST, SSS and MLD for CARIB12, GLORYS12 observational products and CESM-POP. Note that the CESM-POP simulation ends in 2018.

| field | time range | model/dataset | mean | standard deviation |
|-------|-----------|---------------|------|--------------------|
| **SST** | 2000-2020 | CARIB12 | 27.50°C | 0.53°C |
| | | GLORYS12 | 27.63°C | 0.56°C |
| | | OISST | 27.63°C | 0.55°C |
| | 2000-2018 | CESM-POP | 27.47°C | 0.52°C |
| **SSS** | 2000-2020 (DJF) | CARIB12 | 35.86 PSU | 0.94 PSU |
| | | GLORYS12 | 35.72 PSU | 0.91 PSU |
| | | MOGO | 35.82 PSU | 0.45 PSU |
| | 2000-2018 (DJF) | CESM-POP | 36.42 PSU | 0.54 PSU |
| **SSS** | 2000-2020 (JJA) | CARIB12 | 35.35 PSU | 1.69 PSU |
| | | GLORYS12 | 35.19 PSU | 1.77 PSU |
| | | MOGO | 35.49 PSU | 0.96 PSU |
| | 2000-2018 (JJA) | CESM-POP | 36.25 PSU | 0.76 PSU |
| **MLD** | 2000-2020 (DJF) | CARIB12 | 37.90 m | 12.56 m |
| | | GLORYS12 | 35.42 m | 11.11 m |
| | | Deboyer | 41.80 m | 10.52 m |
| | 2000-2018 (DJF) | CESM-POP | 66.22 m | 17.46 m |
| **MLD** | 2000-2020 (JJA) | CARIB12 | 17.49 m | 7.21 m |
| | | GLORYS12 | 17.55 m | 4.72 m |
| | | Deboyer | 23.33 m | 8.31 m |
| | 2000-2018 (JJA) | CESM-POP | 30.45 m | 8.82 m |





**Table 4.** Mean ocean transports (Sv) across passages in the Caribbean Sea: estimates from CARIB12 and the GLORYS12 reanalysis are for the period 2000-2020; estimates from observations are for the date ranges indicated in the table and are included together with relevant references. The location of each passage is shown in Figure **??**. Negative transports correspond to a westward or southward flow, and positive transports to eastward or northward flows.

| Passage | CARIB12 | GLORYS | Observations [date range] | Observations reference |
|---|---|---|---|---|
| Windward Passage | -1.91 | -2.8 | -3.8/-3.6 [Oct '03–Feb '05] | Smith et al. (2007) |
| Mona Passage | -0.38 | +1.22 | -3.0 [Mar '96, Jul '96] | Johns et al. (2002) |
| Anegada Passage | -1.53 | +0.14 | -2.5 $\pm$ 1.4 [spread across the 1990's] | Johns et al. (2002) |
| | | | -4.8$\pm$ 0.32 [Oct '20, Jul '21, Sep '21, Mar '22] | Gradone et al. (2023) |
| Antigua Passage | -2.11 | -4.06 | -3.1 $\pm$ 1.5 [spread across the 1990's] | Johns et al. (2002) |
| Guadeloupe Passage | -0.74 | -0.10 | -1.1 $\pm$ 1.1 [spread across the 1990's] | Johns et al. (2002) |
| Dominica Passage | -2.79 | -3.02 | -1.6 $\pm$ 1.2 [spread across the 1990's] | Johns et al. (2002) |
| St. Lucia Passage | -1.97 | -4.92 | -1.5 $\pm$ 2.4 [spread across the 1990's] | Johns et al. (2002) |
| St. Lucia - Trinidad | -9.51 | -16.62 | -8.6 [spread across the 1990's] | Johns et al. (2002) |





**Figure 1.** Model domain and topography. (a) CARIB12 regional domain and dominant processes regulating water properties in the Caribbean Sea. The northwestward flow of the Amazon river plume towards the Caribbean Sea is known to branch off in the three main pathways shown (red arrows). The plume waters can affect the vertical structure and strength of the Caribbean current (blue arrow). Interactions of the flow with the topography along the numerous passages at the Atlantic/Caribbean Sea interface generate mesoscale eddies (magenta) that entrain and transport waters westward. The North Atlantic Gyre dominates the near-surface inflow between the greater Antilles (orange). The deep flows there form an important return branch of the AMOC. White dashed lines show the radius of influence of boundary nudging layers in the configuration. The green box outlines the focus region for the analysis. (b) Validation and analysis region. Arrows indicate mean flow direction across the multiple passages in the eastern Caribbean Sea and between Greater Antilles. Yellow dashed lines show approximate location of WOCE lines A20 and A22 used in the validation.





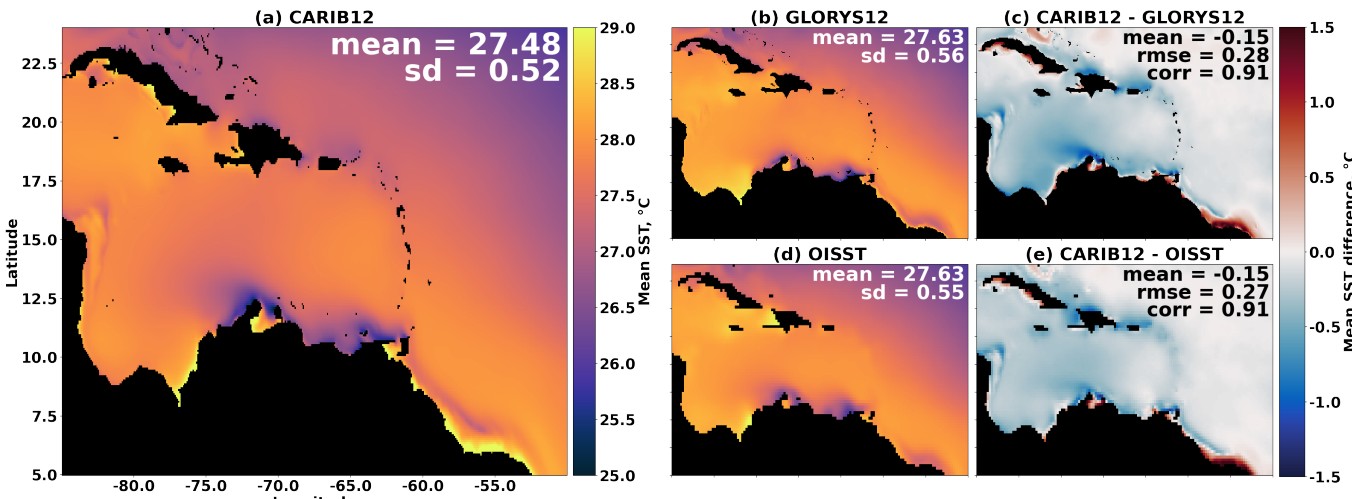

**Figure 2.** Time mean SST (°C) in the region of interest for this study (green box in Figure 1), during 2000-2020: (a) CARIB12, (b) GLORYS12, (c) CARIB12 minus GLORYS12, (d) OISST, (e) CARIB12 minus OISST. Mean and standard deviation are shown for each product (a, b and d). The mean bias, root mean squared error (rmse), and spatial correlation (corr) are shown in each comparison panel (c and e).





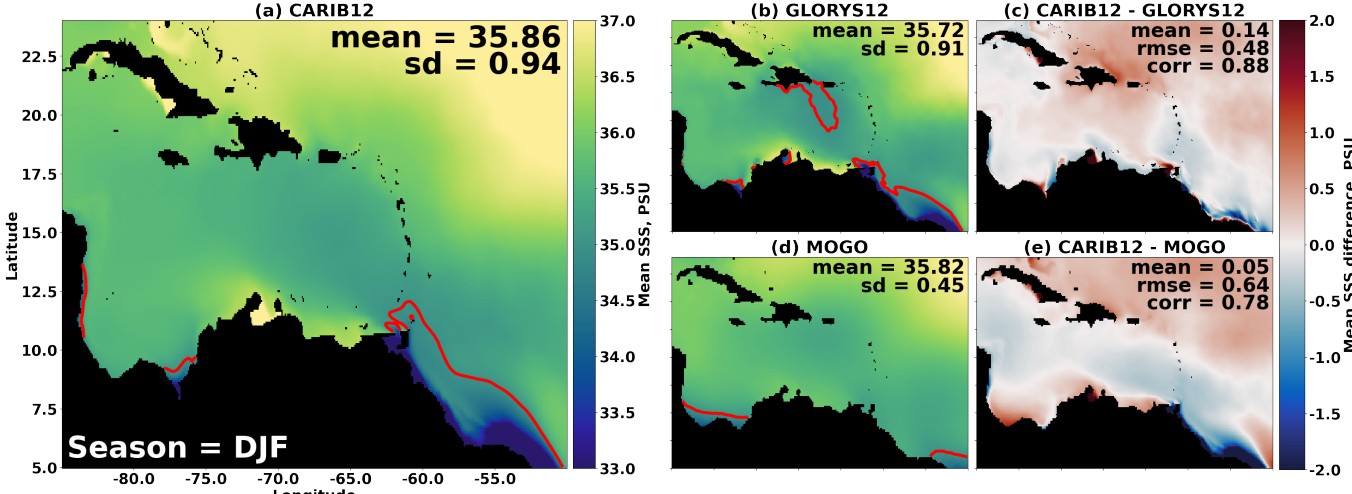

**Figure 3.** Time mean winter SSS (PSU) in the region of interest for this study (green box in Figure 1), during 2000-2020: (a) CARIB12, (b) GLORYS12, (c) CARIB12 minus GLORYS12, (d) MOGO, (e) CARIB12 minus MOGO. The red line in panels (a, b, d) shows the 35 PSU isohaline as a proxy for the extent of the freshwater plumes from the Amazon and Orinoco rivers. Mean and standard deviation are shown for each product (a, b and d). The mean bias, root mean squared error (rmse), and spatial correlation (corr) are shown in each comparison panel (c and e).





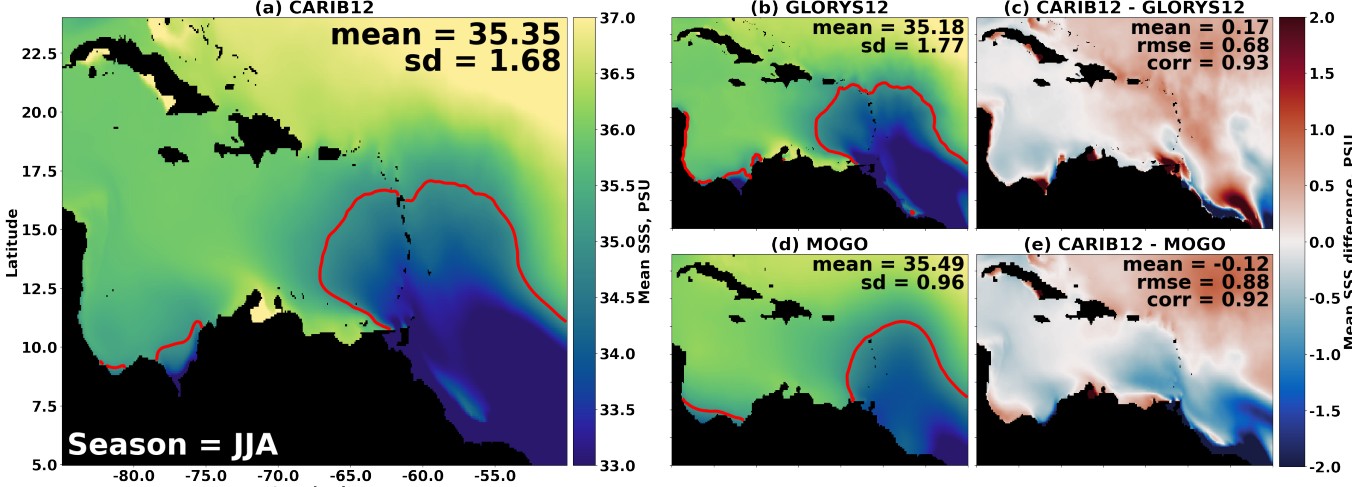

**Figure 4.** Same as Fig. 3, now for SSS during summer (JJA).





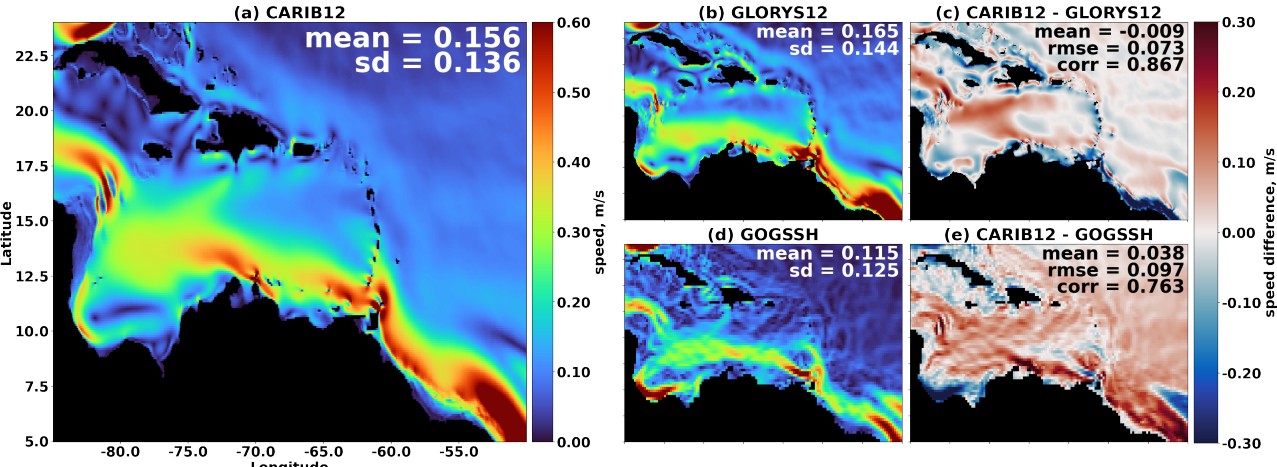

**Figure 5.** Time mean surface speed (m/s) in the region of interest for this study (green box in Figure 1), during 2000-2020: (a) CARIB12, (b) GLORYS12, (c) CARIB12 minus GLORYS12, (d) GOGSSH (altimetry based product), (e) CARIB12 minus GOGSSH. Mean and standard deviation are shown for each product (a, b and d). The mean bias, root mean squared error (rmse), and spatial correlation (corr) are shown in each comparison panel (c and e).




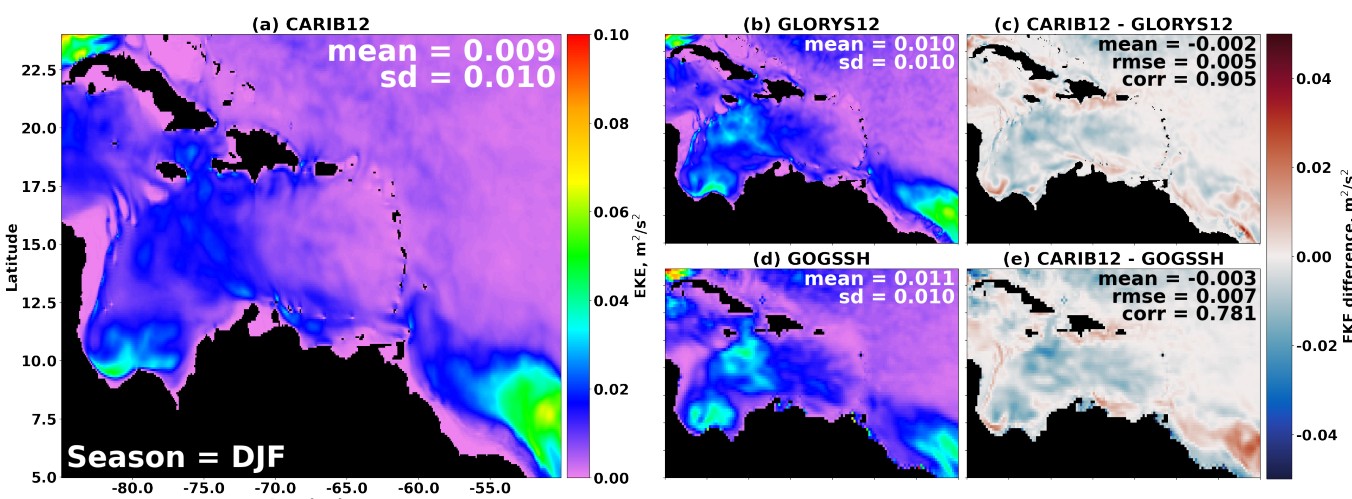

**Figure 6.** Mean winter (DJF) EKE ($\frac{m^2}{s^2}$) in the region of interest for this study (green box in Figure 1), during 2000-2020: (a) CARIB12, (b) GLORYS12, (c) CARIB12 minus GLORYS12. Mean and standard deviation are shown for each product (a, b and d). The mean bias, root mean squared error (rmse), and spatial correlation (corr) are shown in each comparison panel (c and e).





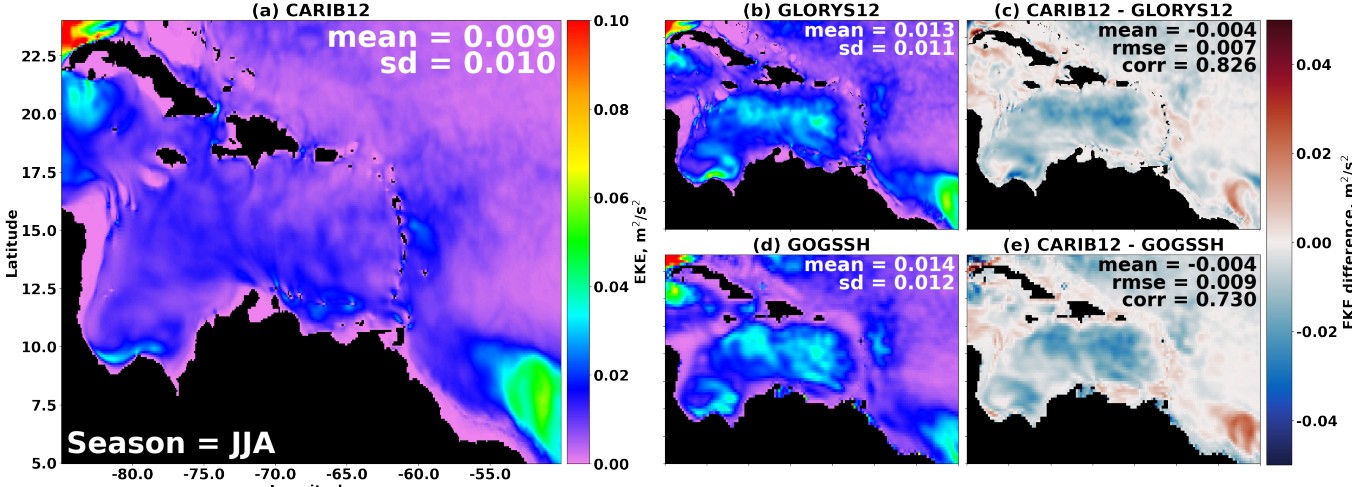

**Figure 7.** Same as Fig. 6, now for EKE during JJA.





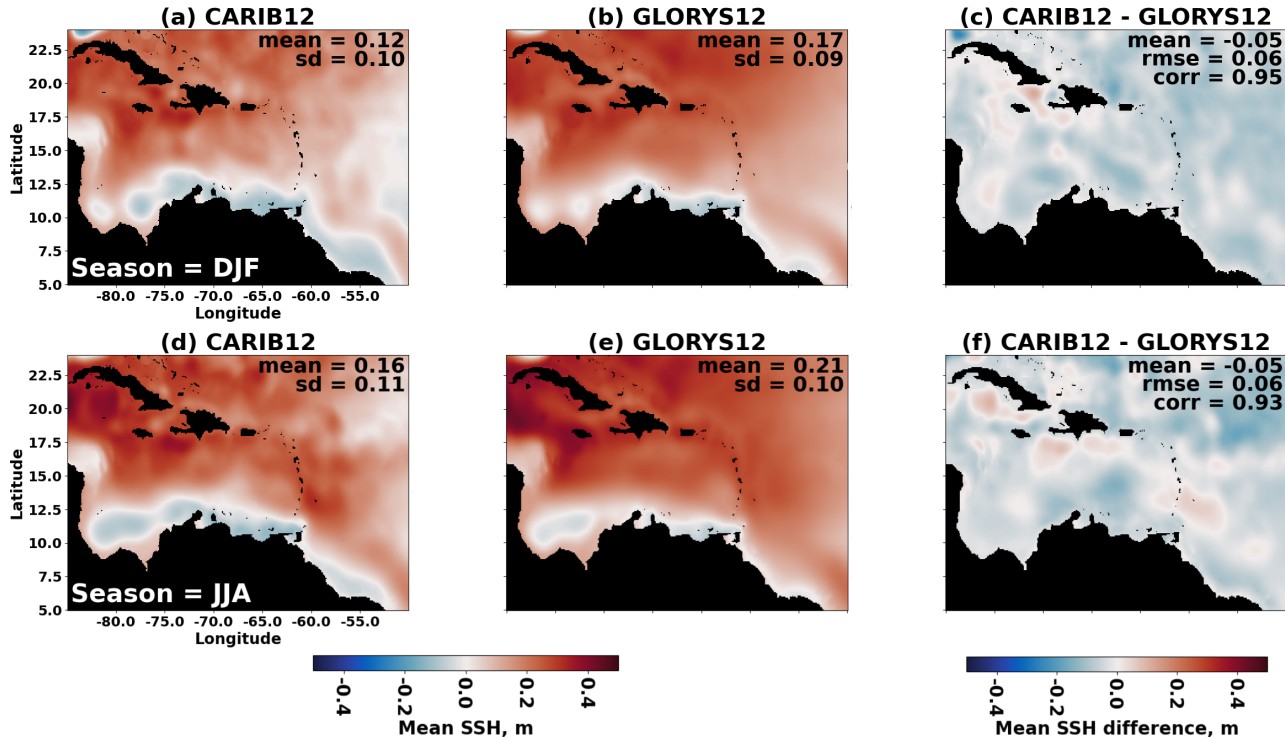

**Figure 8.** Time mean SSH (m) during DJF (panels a-c) and JJA (panels d-f) in the region of interest for this study (green box in Figure 1), during 2000-2020: (a) CARIB12, (b) GLORYS12, (c) CARIB12 minus GLORYS12. Mean and standard deviation are shown for each product (a, b, d and e). The mean bias, root mean squared error (rmse), and spatial correlation (corr) are shown in each comparison panel (c and f).



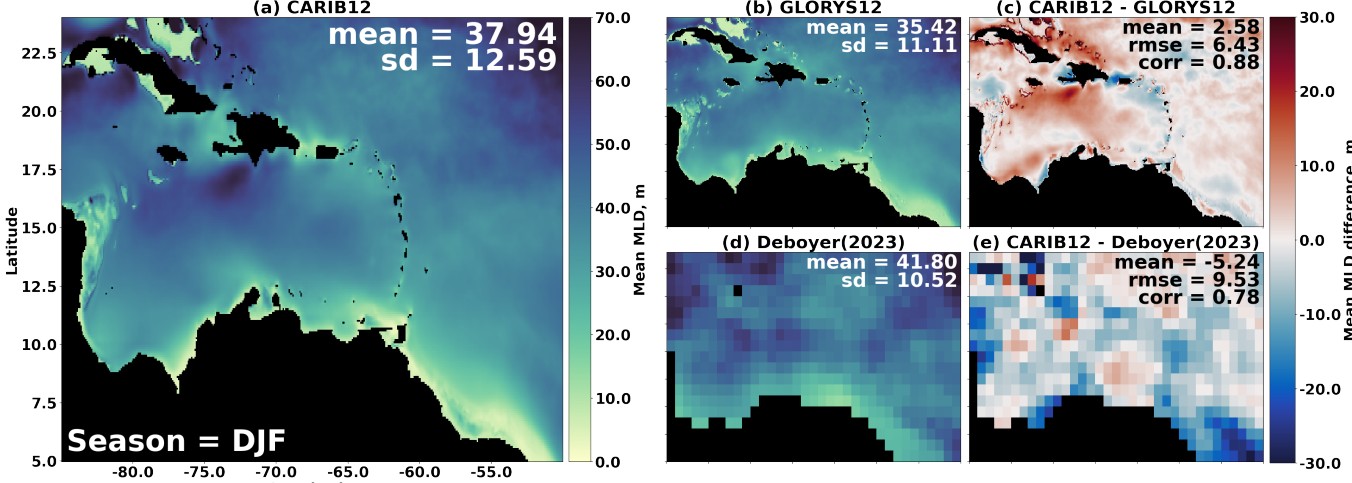

**Figure 9.** Time mean winter (DJF)MLD (m) in the region of interest for this study (green box in Figure 1), during 2000-2020: (a) CARIB12, (b) GLORYS12, (c) CARIB12 minus GLORYS12, (d) Deboyer, (e) CARIB12 minus Deboyer. Mean and standard deviation are shown for each product (a, b and d). The mean bias, root mean squared error (rmse), and spatial correlation (corr) are shown in each comparison panel (c and e).



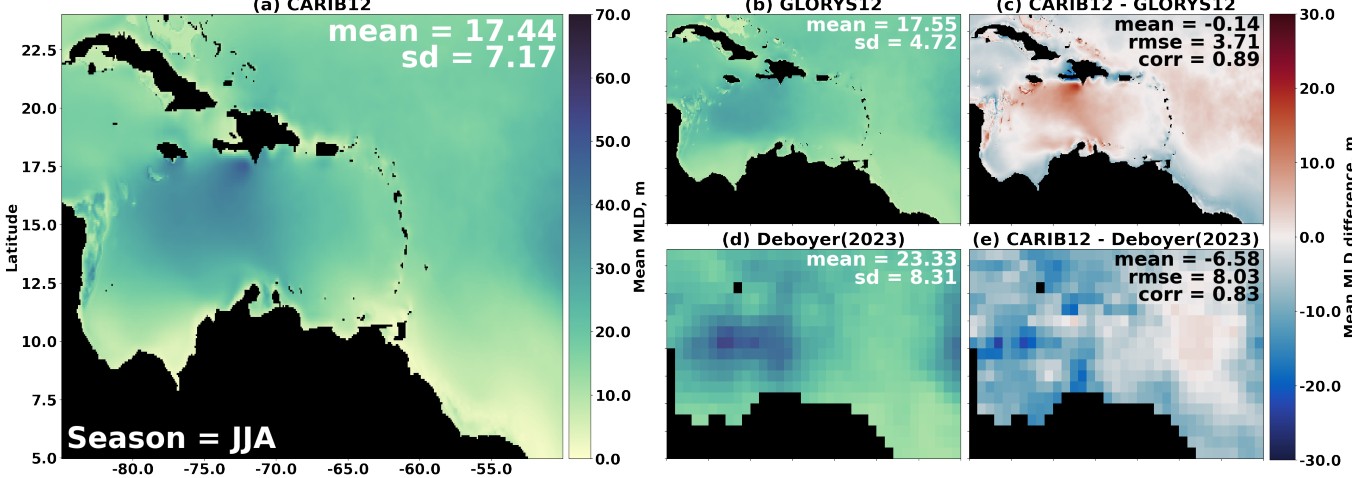

**Figure 10.** Same as Fig. 9, now for MLD during summer (JJA).





**Figure 11.** Temperature-Salinity joint probability density function for (a) CARIB12, (b) GLORYS12 and, (c) GO-SHIP. Panel (d) shows the difference between panel (a) and (b). Panel (e), the difference between (a) and (c).





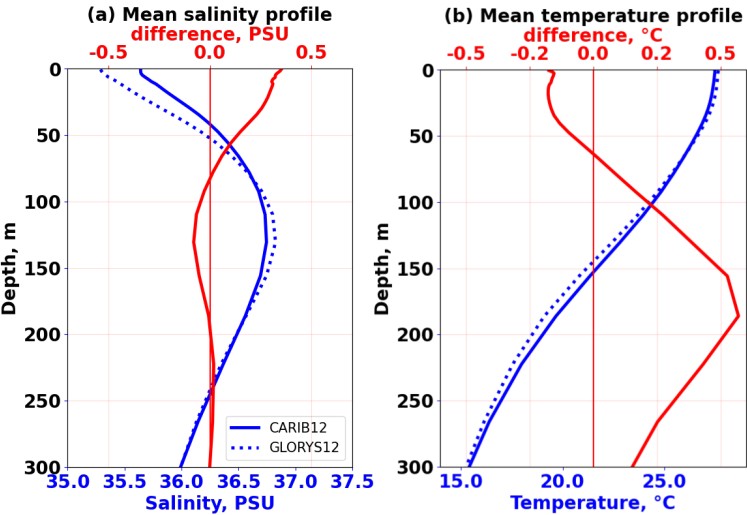

**Figure 12.** Time mean salinity (a) and temperature (b) profiles for the area average within the region of interest for this study (green box in Figure 1): CARIB12 (light blue line) and GLORYS12 (blue). The difference between the two products is shown in red.





**Figure 13.** Seasonal anomalies for salinity (a, b) and temperature (d,e): area weighted average within the region of interest for this study (green box in Figure 1) based on CARIB12 (a, d) and GLORYS (b, e). Differences between the two products are shown in panels (c) and (f) for salinity and temperature, respectively.



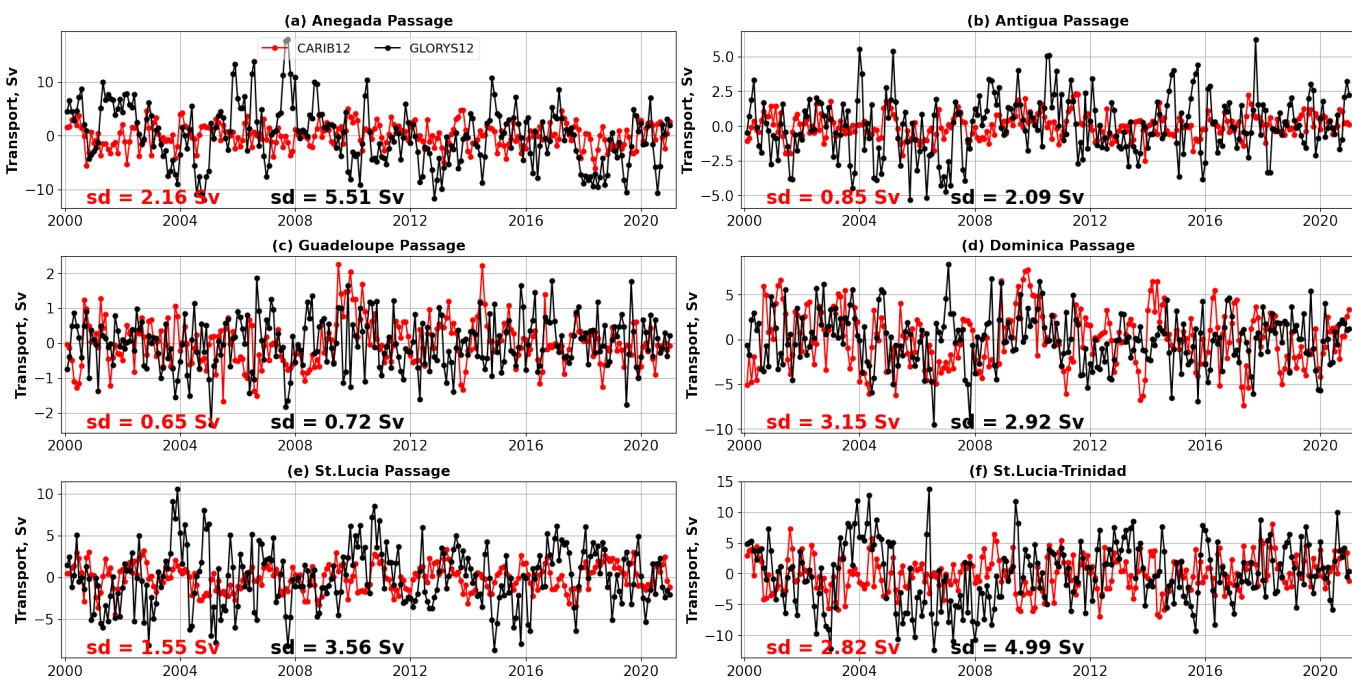

**Figure 14.** Transport through passages in the Eastern Caribbean Sea: monthly anomalies from CARIB12 (red) and GLORYS12 (black) for (a) Anegada Passage, (b) Antigua Passage, (c) Guadeloupe Passage, (d) Dominica Passage, (e) St. Lucia Passage and, (f) the St.Lucia - Trinidad transect. The standard deviation of the time series is indicated at the bottom of each panel, color coded by product. Time mean transports for each passage can be found in Table 4 for both CARIB12 and GLORYS12, as well as available observations.







**Figure 15.** Transport through passages in the Eastern Caribbean Sea from CARIB12 (red) and GLORYS12 (black): (a-f) seasonal anomalies, and (g-l) yearly anomalies. In panels (a-f), the shading shows the standard deviation for each month (i.e. standard deviation of all January means, and so on).

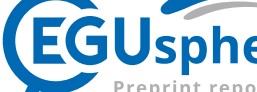

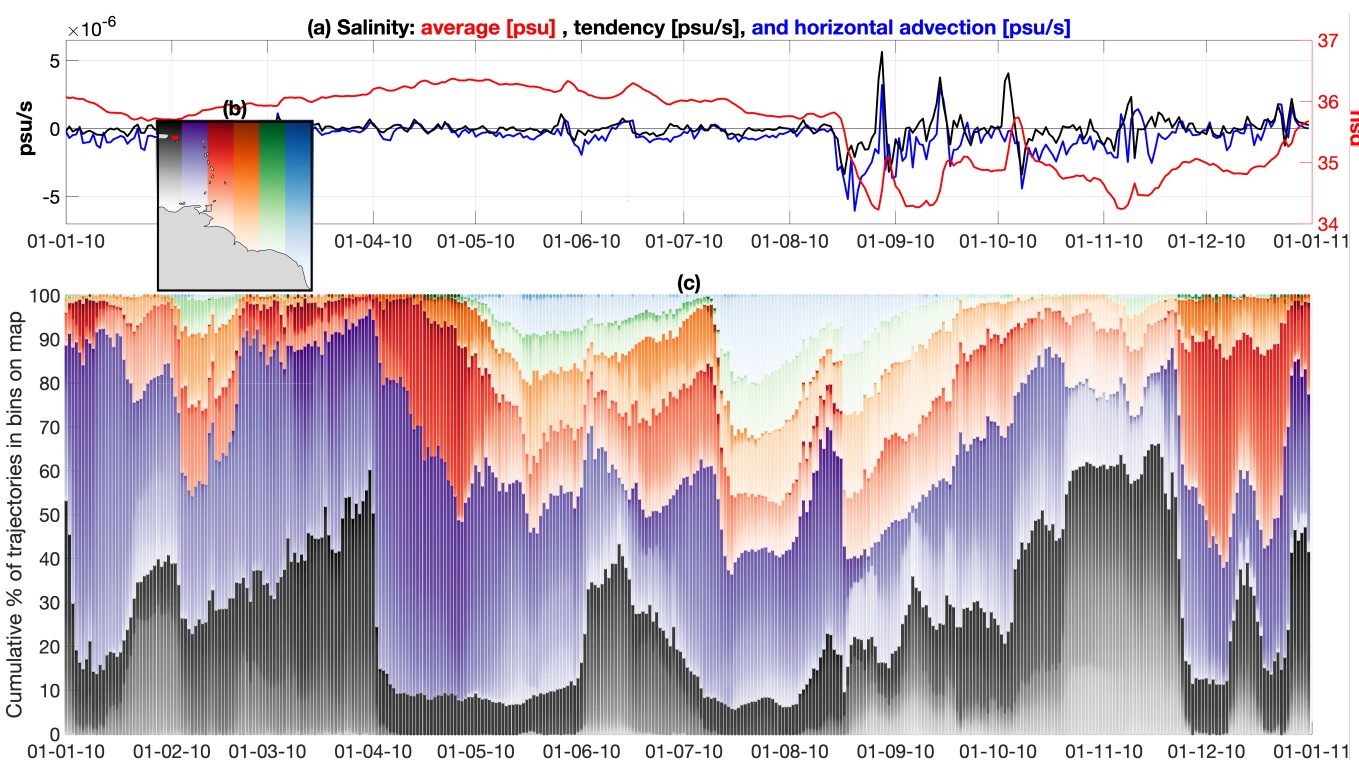

**Figure 16.** (a) Time series of SSS (red, right vertical axis), horizontal SSS advection (blue, left vertical axis) and SSS tendency (black, left vertical axis) within the VIB during 2010. (b) Map with color-coded bins used to identify the path of trajectories in panel (c). A change in bin color denotes a change in longitude and a change in shade (from light to dark) represents a change in latitude (from south to north). (c) Time series of cumulative percent of trajectories in bin: for each particle release date (i.e., for each arrival date into the region of interest, as we are backtracking) we stacked bins following the description in Seijo-Ellis et al. (2023). Bin colors indicate bin locations and correspond to the color distribution in panel (b). As an example, a high cumulative percent of trajectories in light colored bins (red through blue) indicates that particles arrived to the VIB from the south-west, along a path close the coast of South America.



**Figure A1.** Schematic demonstrating the method used to generate Figures 16. a) Example positions of a trajectory for two particles (one black and one white). The 'Release' bin corresponds to the release site from which particles are backtracked. Positions within $0.2° \times 0.2°$ bins are counted and mapped as shown in panel (b). (c) Bins are now color-coded by their location (actual example in the inset map). A change in color represents a meridional position change, while a change in shading corresponds to a latitudinal position change. b) Example of position counting in each bin. (d) Example schematic of how the time series is generated. For each arrival date bins are stacked by the number of particles in each bin (scaled by the number of bins with recorded particles). (e) Actual example of the time series. The colored bins stacked on top of each day in the x-axis represent the pathway the waters came from. The inset map in panel (c) shows the color-coded bins to identify pathways. For a detailed description refer to Seijo-Ellis et al. (2023).



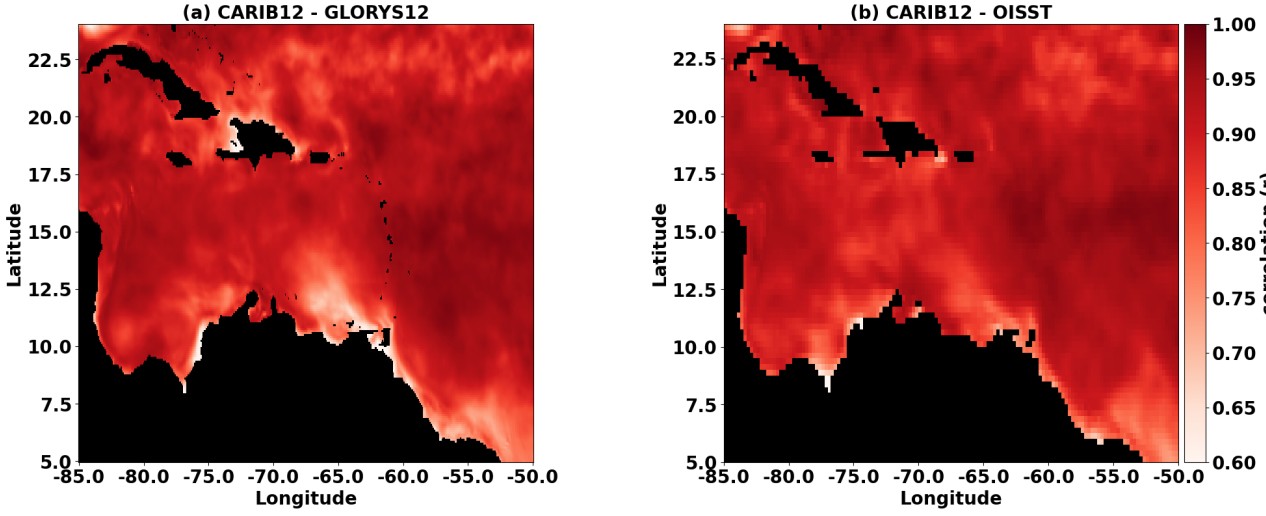

**Figure A2.** Correlation of the 12 month SST climatology between: (a) CARIB12 and GLORYS12, and (b) CARIB12 and OISST.



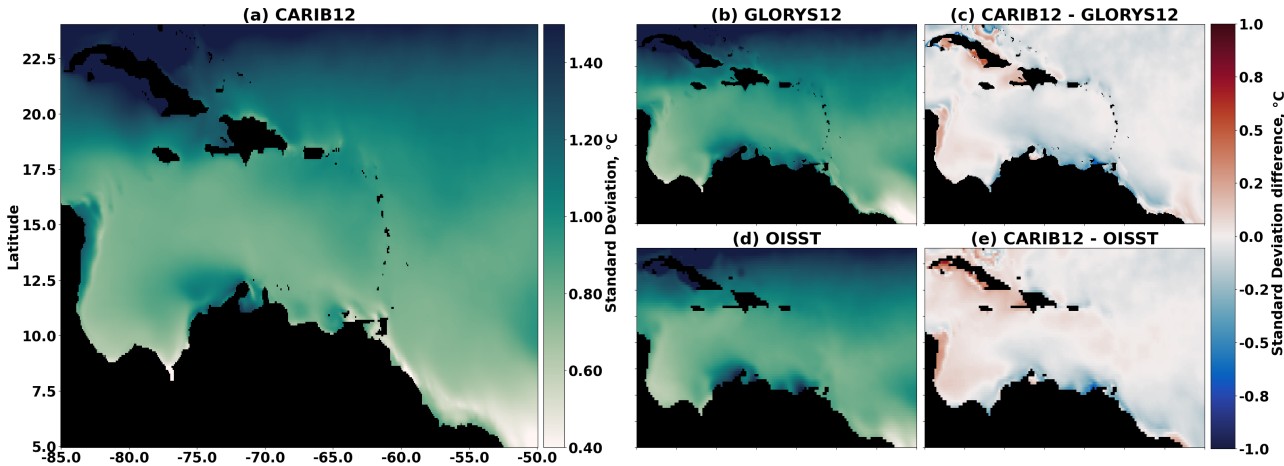

**Figure A3.** Standard deviation of the 12 month SST climatology from (a) CARIB12 (b) GLORYS12 (d) OISST. (c) Difference between the standard deviation of CARIB12 and GLORYS. (d) Difference between the standard deviations of CARIB12 and OISST.





**Figure A4.** Mean fields and metrics for the 1° CESM-POP simulation within region of interest: (a) time mean SST (2000-2018), (b) winter time mean SSS (2000-2018), (c) summer time mean SSS (2000-2018), (d) winter time mean MLD (2000-2018), and (e) summer time mean MLD (2000-2018). The red line in (b) and (c) corresponds to the 35 PSU contour line. Spatial means and standard deviations are specified in the inset text for each panel.



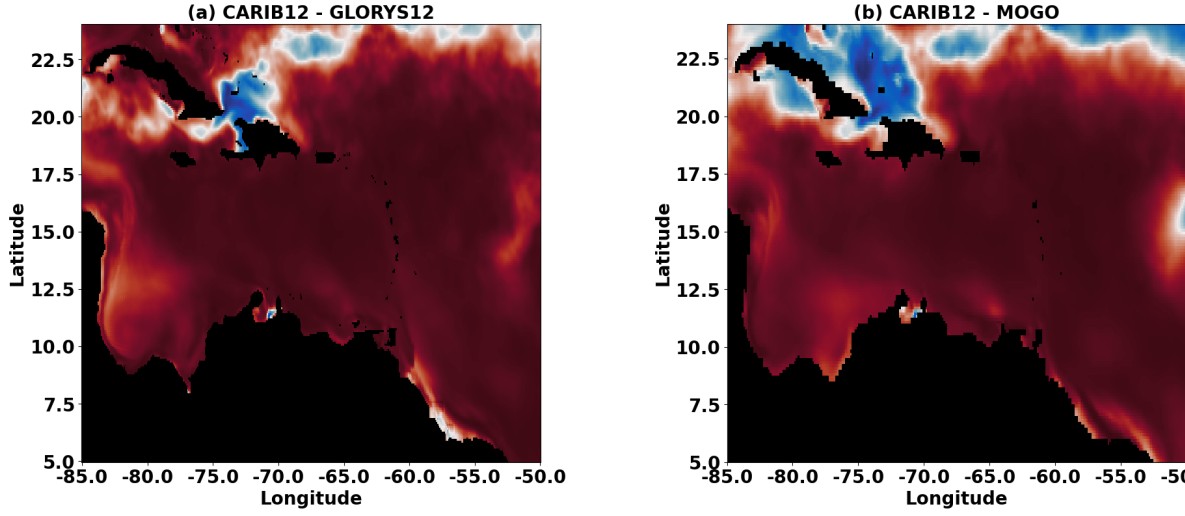

**Figure A5.** Correlation of the 12 month SSS climatology between: (a) CARIB12 and GLORYS12, and (b) CARIB12 and MOGO.



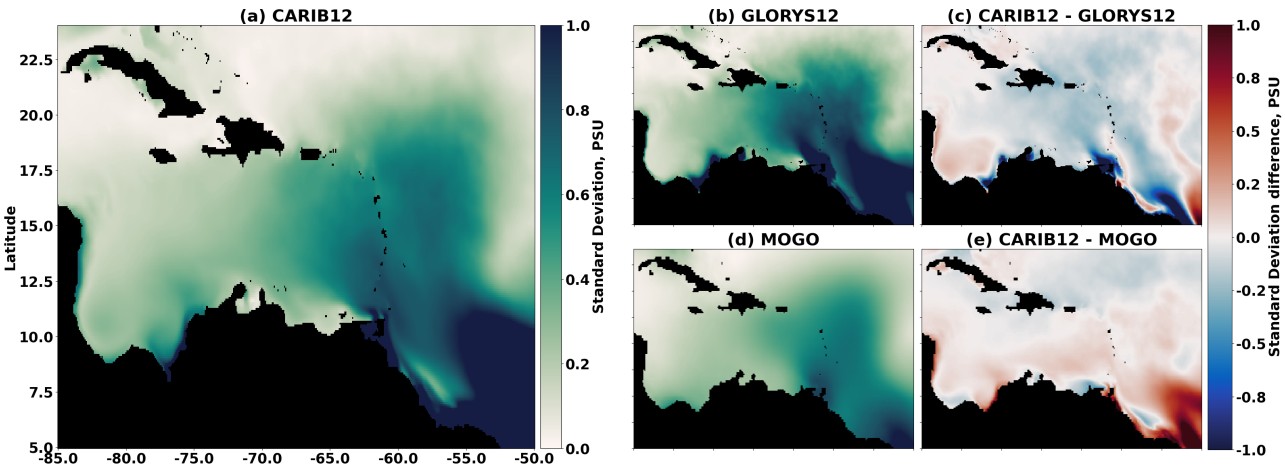

**Figure A6.** Standard deviation of the 12 month SSS climatology from (a) CARIB12 (b) GLORYS12 (d) MOGO. (c) Difference between the standard deviation of CARIB12 and GLORYS. (d) Difference between the standard deviations of CARIB12 and MOGO.



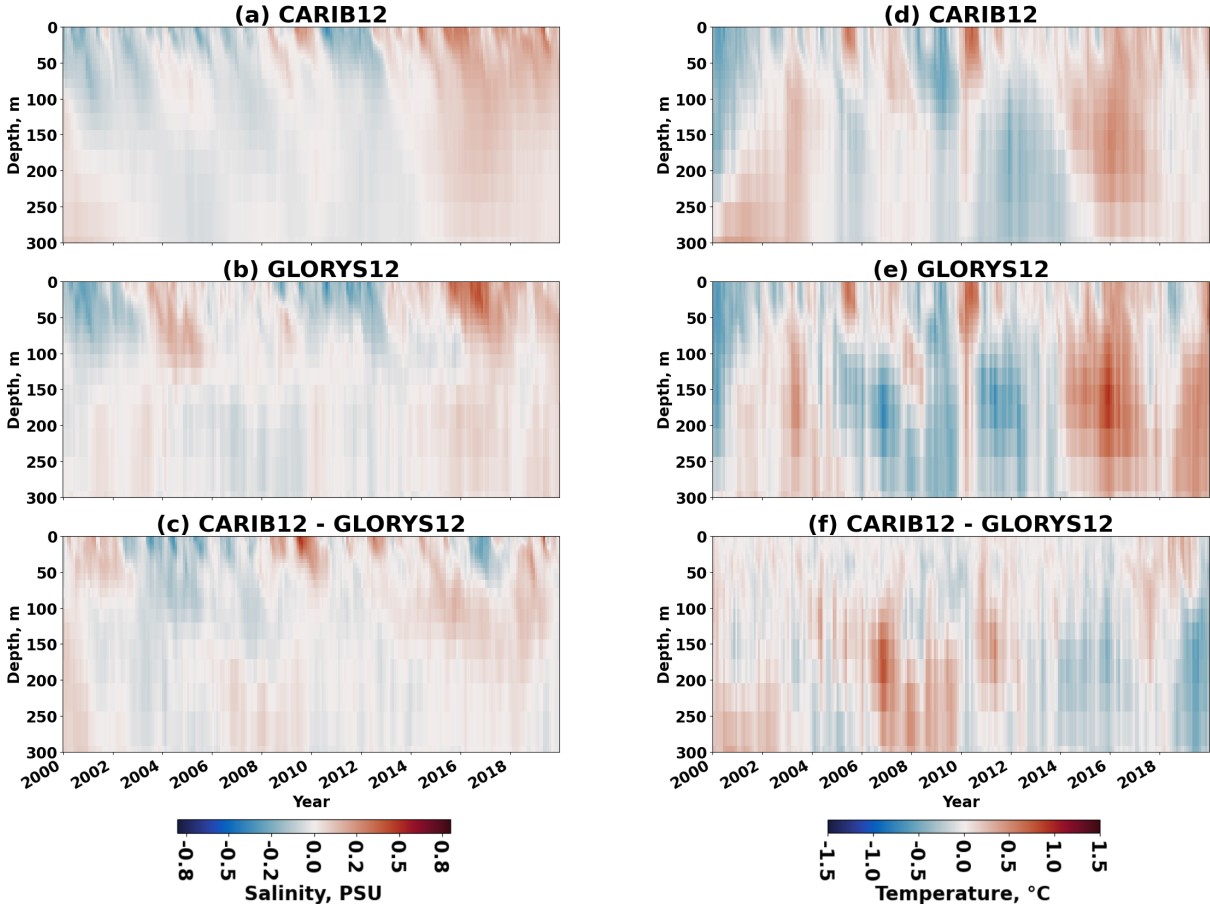

**Figure A7.** Monthly time series of salinity (a-c) and temperature (d-f) anomalies for the full 20-year simulation: area weighted average within the region of interest for this study (green box in Figure 1) based on CARIB12 (a, d) and GLORYS (b, e). Differences between the two products are shown in panels (c) and (f) for salinity and temperature, respectively.