# Peer review of "CARIB12: A Regional Community Earth System Model / Modular Ocean Model 6 Configuration of the Caribbean Sea"

_EGUsphere, 2024_

## Author Comment (AC1)

**Reviewer 2 (reviewer comments in bold font type)**

**This manuscript describes and validates an ocean only CESM/MOM6 configuration of the Caribbean Sea. I think this work has the seeds for a work publishable in GMD but it needs some improvements before it can be accepted for publication. The validation is very basic and it could be substantially improved. Since this is a regional model, one would expect it to perform better than global models given the added physics and the parameter tuning to better fit the regional dynamics. Increased resolution is not a possibility here since the chosen resolution is the same as the model used to force at the boundaries, a choice that is not justified in the text. It seems therefore that one of the benefits in terms of physics is that the model has tides, which are not resolved in the global products, and some extra parameterizations such as that associated with (not resolved) submesoscale variability (just that associated with Mixed Layer Instability). Yet, none of these aspects are discussed in the manuscript. There is no validation of the model tides, and the effect of such variability on the seasonal and subseasonal variability is not provided. The same for the MLI parameterization. In addition, the validation of the model just focused on the seasonal variability, by comparing means and monthly means. This is a very low benchmark given how good models and forcing fields are at present time. There is a need for validation at subseasonal scales. In addition, some information seems unnecessary, such as the comparison with CESM-POP and the drifter section, while some other information is lacking, such as a more robust validation for the sub-seasonal component. A lot of weight is given to GLORYS12 as "the truth". The model has tides, which I'll assume makes the simulation much more realistic and differences are expected, so comparing with GLORYS12 might be misleading. Finally, given that a lot of the emphasis is on the Amazon and Orinoco river plumes, more discussion about this is needed. How realistic is GloFAS? how its seasonal cycle compares with the seasonal cycle derived from observations (such as the Dai dataset)? What is the effect of mixing by tides and the mixing parameterizations on the river plumes?**

We thank the reviewer for their time and feedback on our manuscript. Our goal is to develop a configuration for dynamical downscaling of low resolution climate simulations (typically 0.25º and coarser), we do not aim at downscaling GLORYS12 or similar reanalysis products.  In order to ensure our configuration is suitable for dynamical downscaling of low resolution climate simulations, we developed the configuration using a high resolution product to confirm the proper representation of processes in the ocean. The use of a reanalysis to force our ocean boundaries ensures that the model can be assessed against observations (there would be no expectation of reproducing observed variability if we forced the boundaries using a climate simulation). Also, CARIB12 is not a forecasting model nor a data assimilating model, hence there is no expectation that the model will reproduce observed variability at daily scales and a validation of daily fields is not included. *The*

*validation presented in the manuscript largely focuses on seasonal to interannual time scales since these are the scales relevant for the applications the configuration has been developed for*. Also, our validation is comparable to the validation of other regional MOM6 configurations developed for climate applications and presented in recent publications (e.g. Ross et al. 2023), and it is not a low benchmark. In some cases, our validation includes more metrics, e.g. an analysis of water pathways. Finally, the comparison with CESM-POP is highly relevant to the purposes of CARIB12 as it highlights the added value of using regional models to dynamically downscale low resolution global simulations for regional climate impact applications.

We include GLORYS12 in our validation of MOM6 as it provides additional information and coverage that some of the observational products do not provide (temperature and salinity at depth in all of the domain for example). While GLORYS12 gives us a useful baseline for the comparison we present in the manuscript, additional comparisons are made with observations where possible to balance the fact that GLORYS12 is a reanalysis and will also have biases.

In our effort to improve river runoff representation we tested runoff from two datasets: the JRA55-do dataset (Tsujino et al. 2018) used for the atmospheric forcing and the GloFAS reanalysis (Zsoter et al. 2021). The simulations with JRA55-do runoff exhibited a phase lag in the seasonal cycle of salinity within the Caribbean Sea that was traced back to biases in the runoff. These biases are not present in the GloFAS dataset and using GloFAS corrected the seasonal cycle of salinity in the Caribbean Sea. The GloFAS dataset has been successfully implemented in other regional ocean configurations including that of Ross et al. (2023). A detailed validation of the GloFAS dataset is not within the scope of the manuscript. Finally, an examination of tidal mixing and the different mixing parameterizations is also beyond the scope of this manuscript.

With this initial clarification, the reviewer's comments are included below in bold font type followed by our response.

- Tsujino, H., Urakawa, S., Nakano, H., Small, R. J., Kim, W. M., Yeager, S. G., ... & Yamazaki, D. (2018). JRA-55 based surface dataset for driving ocean–sea-ice models (JRA55-do). Ocean Modelling, 130, 79-139.
- Zsoter, E., Harrigan, S., Barnard, C., Wetterhall, F., Ferrario, I., Mazzetti, C., Alfieri, L., Salamon, P., Prudhomme, C. (2021): River discharge and related forecasted data from the Global Flood Awareness System. v3.1. Copernicus Climate Change Service (C3S) Climate Data Store (CDS). URL: https://cds.climate.copernicus.eu/cdsapp#!/dataset/cems-glofas-forecast
- Ross, A.C. et al, 2023. A high-resolution physical-biogeochemical model for marine resource applications in the Northwest Atlantic (MOM6-COBALT-NWA12 v1. 0). Geoscientific Model Development Discussions, 2023, 1-65.

**1.The model resolution is the same as GLORYS12, why not higher resolution?**

Thanks for the question. CARIB12 has been developed to dynamically downscale low resolution climate simulations. The use of the GLORYS12 reanalysis is for development purposes only, we are not attempting to downscale GLORYS12.

**2. In "layer z∗" what "layer z*" means? is this a hybrid coordinate?**

Apologies for the confusion. No, this is not a hybrid coordinate. The z* or z-star vertical coordinate is a geopotential coordinate. We have reworded line 97 in the original manuscript as follows: " In the vertical, a 65 level z* (z-star, Adcroft and Campin, 2004) grid is used…"

**3. In "baroclinic time step of 800 s", Table 1 says it is 900 s**

We thank the reviewer for pointing out the error and have made the corresponding correction in line 100 of the original manuscript. The baroclinic time step is 900 s as specified in Table 1.

**4. In "The final configuration was largely determined by the mean flows into the CS across the multiple passages between the Caribbean Islands." It seems that the authors "optimized" the model configuration to better represent the inflow at the passages. Given the spatial domain of the model, why is this small area chosen as the benchmark? For instance, in the discussion about surface salinity several deficiencies are acknowledged regarding representation of the river plumes, and in the discussion about surface currents deficiencies in reproducing the mesoscale in the eastern CS is also acknowledged.**

Thanks for your comments. The Caribbean Sea is a relatively large area of our domain and is the main region of interest for our configuration. The inflow through the minor Antilles into the eastern Caribbean Sea is responsible for seasonal to interannual variability and accurately representing these transports is important. In addition, the Caribbean Sea is a significant contributor to the flows into the Gulf of Mexico as well as the northward surface and subsurface ocean mass transports relevant to the overturning circulation. Thus, properly reproducing the flows into the Caribbean Sea is important for reproducing variability in the transport of important physical properties of waters such as heat and salt, as well as variability in the overturning circulation.  As is further described in other comments below and in the manuscript, there are deficiencies that are inherent to limitations in the model (for example the representation of the river plumes), thus during the development of CARIB12 we tuned the model such that the biases resulting from these limitations are greatly reduced within the main regions of interest of the domain (Caribbean Sea in this particular case). For example, as discussed in Section 3.1.1 and shown in

Figures 3 and 4, the salinity biases near the Amazon river delta (resulting from the limitations in the river runoff specification) are drastically reduced within the Caribbean Sea. We now clarify the importance of representing the mean and seasonal flows into the Caribbean Sea in Section 3.4.
* * *
**"The parameterization of Fox-Kemper et al. (2011) is implemented for the re-stratification of the mixed layer by sub-mesoscale eddies with a front length scale of 1500m." in "1500m" a space is needed here and in many other places in the text and tables**

**In "Table 2." should be Table 1**

We thank the reviewer for pointing out these two errors and have made the corresponding corrections in lines 109 and 110 of the original manuscript.
* * *
**Section 2.1.1 Initial and Open Boundary ConditionsIn "The open boundary conditions are specified daily", Table 1 says monthly means are used in the nudging layer, please clarify.**

Thanks for your comment. Yes, the open boundary conditions are specified daily as stated in the manuscript, and the nudging layer is based on monthly means. We refer to a later answer (just after the next comment about tides) for clarifications about the open boundary conditions and the nudging layers.
* * *
**In "Ten tidal constituents are specified at the boundaries.". Given the huge spatial extent of this "regional" application, the direct astronomical forcing might be an important contributor to the tides. Why is the tidal potential force not included? There is a need for a robust validation for the tides in the model. How well are the harmonics at coastal stations and along the altimeter tracks resproduced?**

We appreciate the reviewer for posing this question, as this important information was missing in the previous version of the manuscript. We did include the tidal potential forcing from the same 10 constituents as a body force in the momentum equations throughout the entire area. We accounted for the effects of self-attraction and loading using the scalar approximation (Accad and Pekeris, 1978) with a coefficient of 0.094. We have added these details to Table 1 in the manuscript.

We include below several figures comparing the SSH anomaly from CARIB12 to a number of NOAA tide gauges in the Caribbean Sea (https://tidesandcurrents.noaa.gov/). For clarity, we show only one month for each comparison. We show different months and years for different stations as examples of the model performance across different time periods of the simulation. In general, we see good agreement from the model to the tidal gauges. The correlations are generally above 0.7. We note there are differences in the amplitudes and

these could be attributed to several factors:  1) the model SSH has not been filtered to exclude additional contributors to water elevations such as atmospheric effects, 2) the location of the closest point in the model to the tidal gauges is often one to two grid cells apart (~7-15km) as the tide gauges are on the coastlines, 3) the depth of the closest point in the model  is often drastically different from the depth of the actual tide gauge location (for example, the shelf break in the Caribbean Islands is very short and has steep slopes). Any further validation of tides and/or tidal mixing is beyond the scope of this manuscript and the purposes of this configuration. Because of these limitations and the goal of CARIB12 as a downscaling tool for climate scale applications, we have decided not to include this analysis in the manuscript. We have included some sentences in the Conclusions detailing that a simple comparison with tidal gauges (not included) was completed and showed general good agreement with several tide gauges in the Caribbean Sea and Gulf of Mexico.

[Figure]

**Magueyes**

Validation of the model SSH anomaly with the water elevation anomaly for December 2001 at the Magueyes tide gauge station in the southwest of Puerto Rico.

[Figure]

**San Juan**

Validation of the model SSH anomaly with the water elevation anomaly for May 2002 at the San Juan tide gauge station in the north of Puerto Rico.

[Figure]

**Charlotte Amalie**

Validation of the model SSH anomaly with the water elevation anomaly for June 2003 at the Charlotte Amalie tide gauge station in St. Thomas, U.S.V.I.

[Figure]

**Limetree Bay**

Validation of the model SSH anomaly with the water elevation anomaly for June 2005 at the Limetree Bay tide gauge station in St. Croix, U.S.V.I.

[Figure]

**Pilot's Station East**

Validation of the model SSH anomaly with the water elevation anomaly for May 2005 at the Pilot's Station East tide gauge station south of New Orleans, LA.

- Accad, Y., & Pekeris, C. L. (1978). Solution of the tidal equations for the M2 and S2 tides in the world oceans from a knowledge of the tidal potential alone. Philosophical Transactions of the Royal Society of London. Series A, Mathematical and Physical Sciences, 290(1368), 235-266
* * *
**In "Nudging layers for temperature, salinity and velocities are applied to minimize noise at the boundaries that may contaminate the interior" … "The layers are based on mean monthly fields from GLORYS12." I am a bit confused here. The GLORYS12 fields are specified as daily means at the boundaries, but then nudged to monthly means within the nudging area delineated by the white dashed line in Fig 1, using a nudging strength of 0.3 days? I suspect that this very strong nudging to monthly means will kill most of the daily variability coming in from the boundary forcing. Please clarify.**

Thanks for the opportunity to clarify the difference between nudging the baroclinic flow at the boundaries and the specification of a nudging layer (often referred to as sponge layer). The baroclinic flow nudging timescales are 0.3 day (inflow) and 360 days (outflow) as described in the third row of Table1. This baroclinic flow nudging is applied following Marchesiello et al. (2001) with the Orlanski (1976) radiation scheme to daily fields of u and v velocities.

The nudging *layers* described in row 9 of Table 1, restore T, S, u and v towards a monthly mean within the area delineated by the white dashed line in Fig. 1, with timescales in the order of ~10 days (closest to the boundary) and ~150 days (farthest from the boundary, i.e. at the dashed white line in Figure 1). The "sponge nudging" is not strong enough to eliminate the daily variability from the open boundary forcing, but it is sufficient to reduce

'numerical noise' often generated at the boundaries due to wave reflection and strong horizontal shear. For example, the southern end of our eastern boundary has strong currents in and out of the boundary that are sources of strong horizontal shear (equatorial current and counter-current). Early in the development of CARIB12 we saw that this was resulting in artificial currents that would propagate along the boundary with unrealistic temperature and salinity. After a few months of iteration these artifacts would contaminate the center of the domain. To a lesser extent, the northern boundary also developed similar issues. Thus the need for nudging layers in temperature, salinity and velocities. When we applied the nudging layers we tested different combinations: temperature and salinity only, velocities only, and all (velocities, temperature and salinity). We also tested different length scales and strength of the nudging until we settled in the configuration described in Section 2.1.1 and Table 2. The original manuscript specified the timescale of the nudging layers incorrectly and we have fixed this mistake in row 9 of Table 1.

We have included the following lines at the end of Section 2.1.1: "In general, low resolution climate simulations, like those that will be downscaled using CARIB12, do not have high-frequency mesoscale variability that would be drastically attenuated by the applied nudging. Furthermore, the nudging will help prevent the formation of artificial currents and temperature/salinity signals along the boundaries. Nevertheless, as with any model, the nudging layers might need to be revisited depending on the climate simulation being downscaled". We also included the Marchesiello et al. (2001) reference in the description of the baroclinic flow specification at the open boundaries in row 3 of Table 1.

- Orlanski, I. (1976). A simple boundary condition for unbounded hyperbolic flows. Journal of computational physics, 21(3), 251-269.
- Marchesiello, P., McWilliams, J. C., & Shchepetkin, A. (2001). Open boundary conditions for long-term integration of regional oceanic models. Ocean modelling, 3(1-2), 1-20.
* * *
**In "JRA55-do" why is this forcing used? Has this product been favorably validated for the area when comparing with ERA5 or NCEP?**

The JRA55-do dataset was developed following the Ocean Modeling Intercomparison Protocol (OMIP) and is one of a number of atmospheric forcing datasets used by the international ocean modeling community for the surface forcing of ocean-only simulations. Keeping in mind the purpose of CARIB12, and wanting to be consistent with OMIP protocol, we chose JRA-55 as our atmospheric forcing. The following lines have been added in Section 2.1.2: "The JRA55-do was developed following the Ocean Modeling Intercomparison Project protocol, and is also used for many of the CESM simulations that will be downscaled with CARIB12." As detailed in Section 2.1.2, the dataset is described by Tsujino et al. (2018). JRA55-do improves on the JRA55 reanalysis by applying corrections based on satellite derived fields and other atmospheric reanalyses. The paper describing

the dataset also shows a comparison of different fields against other products. A validation of the atmospheric forcing is beyond the scope of this manuscript.

- Tsujino, H., Urakawa, S., Nakano, H., Small, R. J., Kim, W. M., Yeager, S. G., … Yamazaki, D. (2018). JRA-55 based surface dataset for driving ocean-sea-ice models (JRA55-do). *Ocean Modelling, 130*, 79-139. doi:10.1016/j.ocemod.2018.07.002
* * *
**Section 2.2 Validation datasets*The paper focus mostly on reproducing the seasonal variability. That is a very low benchmark.**

Thanks for the opportunity to clarify this point. As discussed above, the purpose of CARIB12 is to downscale low resolution climate simulations.  A seasonal, to interannual and full-time mean validation is an appropriate validation approach for this and the ultimate use case of CARIB12. CARIB12 was not developed as a forecasting model and there is no expectation that the model will reproduce any daily/sub-seasonal variability. We also note that, while Figures 6 and 7 show the seasonal Eddy Kinetic Energy, EKE is largely related to sub-seasonal/high frequency variability by definition.

Based on this and a suggestion by another reviewer, more monthly and interannual validation has been included in the revised manuscript. Figure A7 has been incorporated in the main manuscript and is no longer part of the Appendix. We also included seasonal and interannual validation of temperature and salinity for subregions of the larger validation area. We have also added the following sentences to the end of Section 2.2: "The overall validation focuses on the seasonal to interannual time scales. As this configuration has been developed for dynamical downscaling of climate simulations, validating seasonal to interannual variability is the target benchmark."
* * *
**In "Optimum Interpolation SST". This is a ¼ deg resolution. There are many other higher resolution datasets. Any particular reason for using this in the validation?**

Thanks for your question. The OISST is commonly used for model validation (Ross et al. 2023 is a good recent example). The dataset incorporates satellite retrievals with ship-based profiles, buoy data as well as Argo profiles. While the product has a resolution of 0.25°, it captures spatial and temporal variability well. Furthermore, Huang et al., (2021) shows that the OISST compares well with high-resolution SST products such as the Group for High Resolution SST (GHRSST) Multiproduct ensemble, with a global difference between OISST and GHRSST of -0.01°C (see Figure 9 of Huang et al., 2021).

The figure below shows a validation using the Operational Sea Surface Temperature and Sea Ice Analysis (OSTIA, Good et al., 2020). This is a 0.05° product that incorporates *in-situ* and satellite data and follows the same GHRSST data processing convention. The mean bias between CARIB12 and OSTIA is -0.18°C, that is a 0.03°C difference from the biases

shown in Figure 2 comparing against OISST. The mean SST, standard deviation as well as the spatial patterns are very similar between OISST and the high resolution OSTIA. Furthermore the biases between CARIB12 and the two products are also very similar. Thus we have decided to retain the comparison with OISST in the manuscript. Please note that we have also included below a new version of the original Figure 2.

[Figure]

New version of Figure 2 in the manuscript but with the OSTIA product instead of OISST.

[Figure]

Revised version of Figure 2 in the manuscript comparing the time mean SST in CARIB12 (a) against GLORYS12 (b-c) and OISST (d-e).

- Ross, A.C. et al, 2023. A high-resolution physical-biogeochemical model for marine resource applications in the Northwest Atlantic (MOM6-COBALT-NWA12 v1. 0). Geoscientific Model Development Discussions, 2023, 1-65.
- Huang, B., C. Liu, V. Banzon, E. Freeman, G. Graham, B. Hankins, T. Smith, and H.-M. Zhang, 2020: Improvements of the Daily Optimum Interpolation Sea Surface Temperature (DOISST) Version 2.1, Journal of Climate, 34, 2923-2939. doi: 10.1175/JCLI-D-20-0166.1

- Good, S.; Fiedler, E.; Mao, C.; Martin, M.J.; Maycock, A.; Reid, R.; Roberts-Jones, J.; Searle, T.; Waters, J.; While, J.; Worsfold, M. The Current Configuration of the OSTIA System for Operational Production of Foundation Sea Surface Temperature and Ice Concentration Analyses. Remote Sens. 2020, 12, 720, doi:10.3390/rs12040720
* * *
**In "Our focus is on time averaged fields, with the average computed for the full time series or specific seasons." Monthly means comparison is not a very challenging metric. Why not provide a more ambitious goal such as spatial maps of correlations in time (and rms) using daily values, which are available for satellite analyses?**

Thanks for your question. CARIB12 was developed to downscale low resolution climate simulations and is not a forecasting model nor a data assimilating model. Hence there is no expectation that daily fields will be properly represented. While we agree that a validation of daily fields is certainly ambitious, it would not be meaningful within the context of CARIB12 and we do not save the full ocean output at daily frequencies. We included text in the revised manuscript to clarify the scope of CARIB12.
* * *
**Section 3.1.1 Temperature and salinityThe figures need some work. The latitude and longitudes in panels b-e need to be included here and in other figures. I know they can be read from the left panel, but that is an unnecessary burden for the reader.**

Thanks for your suggestions. We have modified the figures accordingly and added additional changes based on comments by another reviewer. An example for the new format of the figures is included in the following:

[Figure]

**In "small cold bias of -0.15∘C" (Seijo-Ellis et al., 2024, p. 7) is this the spatial mean? please specify.**

Thanks for your question. Yes, this refers to the spatial mean shown in Figure 2. Lines 191 of the original manuscript now reads: "...with a small cold mean spatial bias of -0.15°C…".

**"Notably, GLORYS12 appears to have a freshwater source in the region of the Dominican Republic and Puerto Rico resulting in positive biases within the CS". Here and in many parts in the validation section, I wonder if the differences could be due to the absence of tides in GLORYS12? There is a strong need to validate the tides and show its effect on the model performance. After all, this is one of the main benefits of a regional application when comparing with global models (besides the increased resolution, which is not the case here).**

As included in a response above, we included a validation against several tidal gauges in the Caribbean Sea. Any further exploration of tidal mixing effects in CARIB12 is beyond the scope of this manuscript. Increased resolution and improved physics are some of the main benefits of CARIB12 given that its intended purpose (as discussed above) is downscaling of low resolution climate simulations.

**In "the spread and extent of the plume waters is similar between CARIB12 and GLORYS12, but is much smaller in the gridded observations (particularly during the winter, Figure 3)." Could you comment on the spatial resolution and limitations of the SSS analysis? Is this a blend of SMOS and SMAP, plus available in-situ? If that is the case, there are well known limitations close to the coast, specially in highly populated areas due to radio frequency interference. Moreover, these products are usually provided with a map of the associated error. How big is the analysis error close to the coast? Maybe the differences you are discussing here fall within the expected error bounds.**

Thanks for your questions. The spatial resolution of the SSS product is ⅛ ° as detailed in Table 2. The product is a blend of SMOS, SMAP and *in-situ* measurements. The differences in the extent of the plume described in Section 3.1.1 is unlikely to be a result of coastal biases in the SSS product (the plume extends far from the coastlines as shown in Figures 3 and 4). Furthermore, the time mean error associated with the MOGO-SSS product in the manuscript is quite small along most of the coastlines (see the scale of the colorbar in the figure below). Based on the reviewer's comment we have added the following line in the description of the validation datasets in Section 2.2: "The errors associated with the Multi Observation Global Ocean Sea Surface Salinity product used here are quite small (in the order of 10^(-3)) even along the coastlines which provides a good baseline for our validation."

[Figure]

MOGO SSS time mean error
* * *
**In "In CARIB12, runoff is not distributed vertically in the ocean but rather spread horizontally across a maximum radius of 600 km with an e-fold decay scale of 200 km at the shallowest layer." This is certainly a very strong limitation. Why is the radius is so big? Did the authors try smaller (i.e., more localized) values? why not? How sensitive is the spread of the river plumes to the vertical mixing parameterization used? This needs to be specified in the text.**

Thanks for your questions. We acknowledge the importance of addressing the limitations in our study regarding the distribution of river discharge. We believe we have been open about these limitations throughout the manuscript. Our selection of the radius (r) and the e-folding scale (e) was determined through trial and error to minimize the bias in sea surface salinity near the Amazon River. We tested various values for r (100, 300, and 600 km) and e (200 and 300 km). The combination of r = 600 km and e = 200 km resulted in the most accurate representation of the Amazon River spreading based on maps of sea surface salinity bias.

Although exploring the sensitivity of river plumes to different vertical mixing parameterizations would be valuable, this is beyond the scope of the current project. The K-Profile-Parameterization (KPP) is used in CESM, which is why this scheme was selected.

In most ocean models, including MOM6, two approaches for representing river discharge are commonly followed. The first approach, currently used in CESM-MOM6, introduces river discharge as "augmented precipitation" over a designated area of the sea surface. The second approach, used in Ross et al., (2023), involves specifying river runoff as a "point source" at the surface and providing an additional source of turbulent kinetic energy to vertically mix the water to a particular depth. Both approaches are ad-hoc and may result in large near-surface salinity biases in coastal regions. We have chosen the first approach because this is the only option available given the vertical mixing scheme used in this study

(KPP). A more desirable approach not presently available in MOM6 would be a dynamically based parameterization, such as the Estuary Box Model (EBM) implemented in the CESM-POP model (Sun et al., 2017). The EBM takes the river discharge and the offshore subsurface salinity from the ocean model as input and delivers the estuarine outflow salinity and net volume flux into and out of the estuary to the ocean model. A project to implement the EBM into MOM6 will begin later this year, and CARIB12 will be one of the benchmarks for evaluating this parameterization.

We added the following sentences after line 134 in Section 2.1.2 of the original manuscript: "The maximum radius for the spreading and the e-fold scale were determined through an ad-hoc process by testing different combinations of these parameters. The combination specified here resulted in improved biases in salinity in the CS, which is our focus."

- Ross, A.C. et al, 2023. A high-resolution physical-biogeochemical model for marine resource applications in the Northwest Atlantic (MOM6-COBALT-NWA12 v1. 0). Geoscientific Model Development Discussions, 2023, 1-65.
- Sun, Q., M. Whitney, F. Bryan and Y.-H. Tseng, 2017. A box model for representing estuarine physical processes in Earth system models. Ocean Modelling, 112, 139-153.
* * *
**Section 3.1.2 Surface currents speed, eddy kinetic energy and sea surface heightIn "The biases are larger when comparing CARIB12 to the altimetry-based GOGSSH (mean bias of 3.8 cm/s), but it is worth noting the difference in spatial resolution and that the altimetry derived velocities are purely geostrophic which may not be a good approximation in this region (surface Ekman currents can reach 50 cm/s in parts of the Caribbean Basin, Andrade-Amaya, 2000)" Is this really true???? The authors do not specify exactly what product from Copernicus they are using, but the only global product I could find specifies: "The total velocity fields are obtained by combining CMEMS satellite Geostrophic surface currents and modeled Ekman currents at the surface and 15m depth (using ERA5 wind stress in REP and ERA5\* in NRT)".**
**The product I am referring to is:**
**https://data.marine.copernicus.eu/product/MULTIOBS_GLO_PHY_MYNRT_015_003/description**

Thanks for your comment and for suggesting a different product. In the revised manuscript, we now use the product suggested by the reviewer for the validation of the model currents. The manuscript has been edited to reflect this change, particularly Sections 2.2 and 3.1.2, as well as Table 2 (which listed the reference for the previous product).  The updated figure with the new dataset is included below.

[Figure]

Please clarify and elaborate on this point in the manuscript.In "Caribbean Sea through the Colombian Basin" Could you provide a lat/lon range for this?

Thank you for your comment and helping make the text clearer. Lines 252-253 now read: "In particular, the increase in EKE extending westward across the Caribbean Sea (65W to 80W) is not as strong in CARIB12 as GLORYS12 and the global currents product suggests."

In "Figure 8 shows the winter and summer mean SSH for CARIB12 and GLORYS12", was the mean SSH (a constant) within the area removed? please specify

Thanks for your question. The original version of Figure 8 included the spatial mean: based on the reviewer's comment, we revised the figure (included below) to remove the area mean. Edits to the text in Sections 2.2 and 3.1.2 have also been included to reflect this change.

[Figure]

**In "The meridional extent of this feature extends further off-shore in CARIB12 indicating a wider Caribbean Current in CARIB12 compared to GLORYS12.", please add the mean currents in both panels to show that is actually the case …**

Thanks for your suggestion. We revised Figure 5 to include the mean currents (see below) so figure speeds and directions are included on the same panels. The meridional extent of the Caribbean Current, notably around 73W, is larger. The magnitude of the speed is also larger. The discussion has been revised with reference to Figure 5 instead of Figure 8 (as numbered in the original manuscript).

[Figure]

A comparison for the SSH is made just for the Mean Dynamic Topography (MDT) and the seasonal values .....the authors could consider using the along track data re-processed for coastal applications
https://www.aviso.altimetry.fr/en/data/products/sea-surface-height-products/regional/x-track-sla/gulf-of-mexico-caribbean-sea.html or
https://www.aviso.altimetry.fr/en/data/products/sea-surface-height-products/global/altimetry-innovative-coastal-approach-product-alticap.html
How are the maps of bias, rms and correlation along the tracks??

Thanks for your question. As discussed in other replies above, CARIB12 has been developed for downscaling low resolution climate applications and a validation of along-track satellite data is beyond the scope of the validation of the configuration. CARIB12 is not a forecasting model nor a data assimilating model, hence there is no expectation that the model instantaneous fields will match along-track satellite data.

In "using the $\Delta 0.03 kg/m3$ density criterion with respect to surface values in CARIB12

**and GLORYS12, and with respect to a depth of 10 m in the deBoyer Montégut (2023) climatology.", why not use the same definition for both cases (10 m reference)?**

Thanks for the question. In the revised manuscript, we use the original MLD climatology of Deboyer et al. (2004), which is calculated with respect to the surface (see figure below). We edited the manuscript to reflect this change, including Section 2.2, Section 3.4 and Table 2. Overall the biases are similar to those reported in the original submitted manuscript.

[Figure]

**In "overall deeper MLD in CARIB12", could this be to the effect of tides on mixing? In "small biases described for salinity", I won't say the biases are small. Fig 4 suggests biases of up to 2 psu!In "an overall positive salinity bias (Figures 3 and 4) corresponds to saltier waters in the near surface which leads to weaker vertical stratification (Figure 12) in the upper 0-100 m resulting in a deeper mixed layer particularly during the winter" I think vertical mixing by tides is another very possible candidate.**

Thanks for your comment. In Lines 272-275 we were referring to the biases within the Caribbean Sea and generally away from the river sources. Biases of up to 2 psu are only present within the the vicinity of the Amazon and Orinoco rivers. In general, across the

Caribbean Sea, the largest biases are less than 1 psu and on average less than 0.2 psu (Figures 3 and 4). We have edited the text on lines 272-275 of the original manuscript to clarify: "The biases within the Caribbean Sea and away from the vicinity of the Amazon and Orinoco rivers…" We have also added a sentence at the end of line 275 of the original manuscript that reads: "Another likely contributor could be the added effect of tides in CARIB12 driving vertical mixing, a process that is not included in GLORYS12".
* * *
**In "one must exercise caution in this comparison as the CARIB12 MLD is calculated using the Δ0.03kg/m3 criterion referenced to the shallowest layer in the model (1.25 m), while the deBoyer Montégut (2023) dataset is calculated referenced to 10 m 285 depth",  so, why not use 10 m instead?**

Thanks for your comment. As described in a reply above (see previous figure), we now use the original MLD climatology of Deboyer et al. (2004), which is calculated with respect to the surface.
* * *
**Section 3.3 Vertical structure and water mass properties**

**This section needs considerable work. The authors mention  that they "identified WOCE lines within the region of interest and specific cruises within the time frame of the simulation (2000-2020)". But here just the WOCE lines are discussed. In addition, the validation is reduced to a visual comparison of the PDFs as a function of depth (and the difference of the PDFs). Given the latitudinal extent of the WOCE lines I suppose there is substantial spatial variability from north to south, yet none of that is considered in the validation. And why not include all the Argo profiles available for the area in the validation of the model. A much more robust validation will be to provide correlation, bias and rms error as a function of depth for all the available profiles (WOCE lines, Argo floats, cruises). The same could be done along the WOCE transects to show where in latitude the deficiencies are.**

Thanks for your comment. Multiple different cruises have taken measurements across each WOCE line: we use cruises that fall within our domain and within the time period of the simulation (2000-2020). All the data available are included in the analysis presented.

The probability density functions (PDFs) shown in Figure 11 are not a function of depth, they are PDFs of the Temperature-Salinity diagrams and are included to validate how the model represents water masses in the domain of interest. As discussed in other comments, comparing instantaneous profiles in the model with observations is beyond the scope of CARIB12.  Also, the full depth model output is not saved at daily frequency (this would require a significant amount of memory storage and is generally not useful for the applications CARIB12 has been developed for), we only save monthly means of temperature and salinity with depth. In the revised manuscript, we provide more information about the ship-based data and clarify the importance of validating how water masses are

represented in CARIB12.
* * *
**In "observations along WOCE lines A20 and A22 within the region of interest", there is a need for more information about this. How many occupations of the two lines between 2000-2020? are these transects during the summer, winter, both? ... If more than one season, does it make sense to combine them all in the same plot?**

Thank you for the suggestion. We have added additional information regarding the WOCE lines in Section 2.2: "A total of 250 profiles from the following cruises are used in the analysis presented here: 43 profiles from 35A3200304, 43 profiles from 316N200309, 71 profiles from 316N200310, 43 profiles from 33AT20120324, and 50 profiles from 33AT20120419."

Each cruise has occupations across different months and years as detailed in the ID tag for each cruise: April 2003, September 2003, October 2003, March 2012 and April 2012, respectively. We include below versions of Figure 11 of the original manuscript for Fall and Spring (based on the timing of the cruises used in the analysis) which are consistent with the original figure in the manuscript. Hence, for the validation of the overall representation of the water masses in CARIB12 we keep the original plot, which combines different cruises, as it incorporates more profiles.

[Figure]

TS diagram with profiles from cruises 316N200309 (September 2003) and 316N200310 (October 2010).

[Figure]

TS diagram with profiles from cruises 35A3200304 (April 2003), 33AT20120324 (March 2012), and 33AT20120419 (April 2012).
* * *
**In "some of the largest differences occurring in the shallower Caribbean surface waters (Figure 11d-e)", how is the reader supposed to know this mismatch is for the shallow areas? It will be better to show a comparison between the model and obs along one of the transects. Or even better, if there are several occupation of a transects, show a transect of the mean difference**

Apologies for the confusion. Line 298 of the original manuscript and in particular "..some of the largest differences occurring in the shallower Caribbean surface waters.." does not refer to the depth of the ocean: "shallower" is used to describe the Caribbean surface water mass which is the shallowest water mass in the Caribbean Sea. The differences are not surprising because surface waters are highly variable due to atmospheric fluxes and other processes

that do not directly affect the lower water masses. We have eliminated the word "shallower" from line 298 to avoid confusion.

As discussed in other comments, while validating water mass representation is helpful, comparing instantaneous profiles in the model with observations is beyond the scope of CARIB12.
* * *
**In "Additional biases are noted around the salinity maximum", again, how is the reader supposed to see that in figure 11?. A spatial map would be much more informative.**

Thanks for your comment. As discussed above, Figure 11 shows PDFs of the Temperature-Salinity diagrams. The x-axis on Figure 11 is the salinity, while the y-axis is the temperature. The salinity maximum occurs on the farthest extension to the right in each panel of Figure 11 a-c. The difference in the PDF is shown in panel d (CARIB12 - GLORYS12) and panel e (CARIB12-GOSHIP). Temperature-Salinity diagrams are useful to understand the overall distribution and characteristics of water masses in the ocean which is the purpose of this manuscript section. In the particular case of our validation, Figure 11 is a useful validation as it shows that the representation of water masses in CARIB12 is realistic. There is no expectation that each individual profile of temperature and salinity between CARIB12 and GOSHIP will match. The PDF on the other hand gives an understanding of the distribution of values which is a more useful comparison for this particular case.
* * *
**In "Figure 13" , the caption narrative is very confusing, not just for this figure. Please revise, and then ask a colleague not familiar with the work if it makes sense. Indicate the state variable in the title, such as CARIB12 temperature, GLORYS12 salinity, etc.**

Thanks for your suggestions. We have included the variable names at the top of each column (new figure attached below). We have also reviewed the Figure caption which now reads as follows: "Area weighted mean seasonal anomalies of salinity (a, b) and temperature (d,e) within the validation region (green box in Figure 1). Panels (a, d) correspond to CARIB12 output and panels (b, e) to the GLORYS12 reanalysis. Differences between the two products are shown in panels (c) and (f)." . We hope these changes make the figure easier to understand.

[Figure]

Revised version of Figure 13 of the original manuscript.
* * *
**In "These results highlight CARIB12's ability to correctly represent temperature and salinity variability in the CS as seasonal scales", why these results show CARIB12 is better??? This is a model to model comparison, and in fact GLORYS12 has in-situ profiles of temperature and salinity assimilated, so in what sense CARIB12 is better????**

Thanks for the question. We clarify in the revised manuscript that the purpose of this configuration is downscaling low resolution climate simulations, not GLORYS12. Hence, our goal is not to improve GLORYS12, which is a data assimilating model. We use GLORYS12 as a baseline and show that CARIB12 compares well against GLORYS12, we do not state that CARIB12 is better than GLORYS12.
* * *
**In "We note that while transports based on CARIB12 and GLORYS12 are a time mean**

**during 2000-2020, observational estimates are available only for shorter periods of time." … "The mean transport across the Windward Passage between Cuba and Hispaniola is 1.91 Sv into the Caribbean Sea which is lower than that in GLORYS12 (2.8 Sv) and observational based estimates (3.8/3.6 Sv) by Smith et al. (2007)": what is the minimal cross section area along this passage in the raw bathymetry vs that in INDO12 and GLORYS12? And also,how was the transport in Smith computed? Check in the model simulation for the assumptions made in Smith, such as extrapolation to the bottom and boundaries? Based on the model simulations, are these valid assumptions? What happens with the transport estimate if you use Smith assumptions? What is the temporal coverage in the observations used to estimate the transport? Does the model suggest substantial seasonal variability that was not captured by the observational period? Check these issues for all passages when comparing with observed estimates.**

Thanks for the comments. When we created the topography for CARIB12, we manually verified the width and depth of each of the main passages along the Caribbean Sea. We have attached an example of what the bathymetry looks like in CARIB12 compared to the source data (SRTM, as detailed in 2.1) along the Windward passage. As detailed in Section 2.1 we generate the topography from the Shuttle Radar Topography Mission (SRTM) 15+V2. (Tozer et al. (2019)) using a Cressman weighted interpolation scheme. We then proceeded to map the topography along each of the passages within the Caribbean Sea and compare transects between the original data and the smoothed data. We then performed manual edits to the smoothed topography to ensure proper representation of the width and depth of each passage. We do not have access to the topography/bathymetry of GLORYS12 and extracting this information from the model output fields is likely not an accurate representation of the actual bathymetry used in GLORYS12. As detailed in Table 4, the transports in Smith et al. (2007) are observation based transports, not modeled transports. This clarification may be helpful for additional questions included in this comment.

The temporal coverage of all the observations is detailed in Table 4. We cannot compare seasonal variability against observations because the data provided in the references are mean transports. Furthermore, most observations take place over a limited period of time (Table 4) that in most cases does not capture seasonal variability. The observational estimates serve simply as a general guide of what the mean transport could be in the real world.

The comparison in Section 3.4 also follows the approach suggested in Griffies et al. (2016), consistent for the validation and standards for OMIP simulations. Much like Table 4 in our manuscript, Table J2 in Griffies et al. (2016) details mean transports across important passages in the world's oceans,  including their source and time frame of the observations. Their standard includes the same Smith et al. (2007) estimate for the Windward Passage that we use in our manuscript. These transports help us to determine whether the mass

flows in the model are captured with some degree of similarity to what the limited observations suggest.

[Figure]

Top left: topography in the region surrounding the windward passage from the SRTM dataset. Top right: topography as originally generated for CARIB12. Bottom: Example of a single transect across the Windward passage (blue line shows the transect as represented in the original CARIB12 topography and light blue shows the transect from the SRTM dataset).

[Figure]

The figure above shows an example of changes manually made to the original topography used in CARIB12. The light blue shows a transect in the area of the Windward passage as represented in the SRTM dataset. The blue line shows the original topography generated for CARIB12, and the red line shows the topography with manual edits along the transect.

- Griffies, S. M., Danabasoglu, G., Durack, P. J., Adcroft, A. J., Balaji, V., Böning, C. W., Chassignet, E. P., Curchitser, E., Deshayes, J., Drange, H., Fox-Kemper, B., Gleckler, P. J., Gregory, J. M., Haak, H., Hallberg, R. W., Heimbach, P., Hewitt, H. T., Holland, D. M., Ilyina, T., Jungclaus, J. H., Komuro, Y., Krasting, J. P., Large, W. G., Marsland, S. J., Masina, S., McDougall, T. J., Nurser, A. J. G., Orr, J. C., Pirani, A., Qiao, F., Stouffer, R. J., Taylor, K. E., Treguier, A. M., Tsujino, H., Uotila, P., Valdivieso, M., Wang, Q., Winton, M., and Yeager, S. G.: OMIP contribution to CMIP6: experimental and diagnostic protocol for the physical component of the Ocean Model Intercomparison Project, Geosci. Model Dev., 9, 3231–3296, https://doi.org/10.5194/gmd-9-3231-2016, 2016.
* * *
**Typo in "(Figure ??)"**

Thank you for pointing out this error. Line 354 of the original manuscript now reads: "The St. Lucia - Trinidad section (Figure 1) is a combination of the St. Vincent…".
* * *
**In "we designed an experiment similar to that in Seijo-Ellis et al. (2023)." If the authors decide to keep this section, a very high percentage of potential readers will not know what was done in Seijo-Ellis et al. (2023). Please briefly describe what is reported in the next paragraph.**

We have decided to move this Figure to the Appendix following this and other comments by both reviewers. The revised Appendix expands the description of what was done in Seijo-Ellis et al. (2023) and why it is relevant to the validation of CARIB12. More about this is also included below, in response to the last comment by the reviewer.
* * *
**In "Figure 16a shows that the seasonal decrease in near-surface salinity is largely driven by near-surface horizontal salinity advection", why just 1 year is shown? How do we know the same applies for other years? Maybe show the mean and standard deviation similar to Fig 15a?**

Thanks for your comment. We moved this section to the Appendix and include an example figure to show that CARIB12 reproduces pathways of Amazon river plumes shown in Seijo-Ellis et al. (2023).

**In "Variability in salinity advection is associated with intrusions of Amazon river waters: salinity starts decreasing between May and June, as Amazon river plume waters arrive into the VIB (as indicated by the light blue colors in Figure 16c)." Now here, out of the blue, the author is talking about drifter trajectories. This is VERY confusing without knowing what was done in Seijo-Ellis et al. (2023). And all this to conclude that the results are consistent with that work. Why replicate work that was done and reported previously? All this space could be used to provide a more convincing model validation for tidal variability and sub-seasonal scales for instance.**

Thanks for your comment. We agree this section feels somewhat disjointed from the rest of the validation in the manuscript, hence we moved it to the Appendix, together with the description of the method used to generate the relevant Figure). We also added text to the main manuscript to clarify the importance to test CARIB12's ability to reproduce known pathways of plume water intrusions into the Virgin Islands basin. While the comparison is not direct (like the validation of the surface fields for example), our results show that indeed CARIB12 is capable of reproducing these pathways. This is important because these pathways are a result of the interactions between the density front of the plume and the topography as the waters approach the eastern Caribbean Sea. Thus, reproducing the pathways indicates that CESM-MOM6 is able to reproduce these processes, with important improvements over coarse resolution climate simulations. In place of this section, we included further analysis of subregions within the Caribbean Sea (as suggested by another reviewer). Further validation of tidal mixing or sub-seasonal variability is beyond the scope of the current work, as described earlier in this document.

---

## Author Comment (AC2)

**Reviewer 1**

We would like to thank the reviewer for their helpful comments and feedback on our manuscript. Below are the reviewer comments in bold text followed by our response.
* * *
**1. Yucatan transport: You estimated a Yucatan transport of 20.6 Sv. It is important to include that value in Table 4 and compare with more recent estimated for the region. Although you cited a couple of studies that seems to be in good agreement with your model result (Sheinbaum et al. [2002] and Candela et al. [2003] with 23.08 and 23.06 Sv, respectively), a more recent study by Candela et al. (2019; DOI: 10.1175/JPO-D-18-0189.1) using an extended time series reported a Yucatan transport of about 27.6 Sv. The observed value is somewhat below the transport you can derive from GLORYS, which is about 29 Sv. This implies that CARIB12 underestimated the Yucatan transport by 25% or more. You should recognize that model bias, and maybe provide some discussion about what could be the reason of this underestimation.**

We thank the reviewer for sharing an updated reference for the transport across the Yucatan channel. We have included the Yucatan transport on Table 4 (shown below) along with the updated reference. We also expanded our discussion in response to this comment (Lines 351-356 in the original manuscript) as follows:

"The net mean inflow to the Caribbean Sea is 20.94 Sv which is 0.31 Sv more than the mean flow out of the Caribbean Sea via the Yucatan Channel (20.63 Sv) (Figure 1a). The mean flow through the Yucatan Channel in CARIB12 is ~3 Sv less than that estimated from observations by Sheinbaum et al. (2002) and Candela et al. (2003) (23.8 and 23.06 SV, respectively); it is also ~7 Sv below GLORYS12 and the most recent estimate of 27.6 Sv by Candela et al. 2019. The difference in mean transport across observational estimates collected in different years may indicate low frequency variability that is not well-captured in CARIB12 and is partially captured in GLORYS12. Also, the estimates by Sheinbaum et al. (2002) and Candela et al. (2003) are based on observations between September 1999 and June 2001, whereas the estimate in Candela et al. 2019 is based on observations between September 2012 and August 2016. We note that the section defining the Yucatan Channel in CARIB12 is not completely bounded by land which may result in a lower mean outflow there."

**Table 4.** Mean ocean transports (Sv) across passages in the Caribbean Sea: estimates from CARIB12 and the GLORYS12 reanalysis are for the period 2000-2020; estimates from observations are for the date ranges indicated in the table and are included together with relevant references. The location of each passage is shown in Figure 1. Negative transports correspond to a westward or southward flow, and positive transports to eastward or northward flows.

| Passage | CARIB12 | GLORYS | Observations [date range] | Observations reference |
|---|---|---|---|---|
| Windward Passage | -1.91 | -2.8 | -3.8/-3.6 [Oct '03–Feb '05] | Smith et al. (2007) |
| Mona Passage | -0.38 | +1.22 | -3.0 [Mar '96, Jul '96] | Johns et al. (2002) |
| Anegada Passage | -1.53 | +0.14 | -2.5 ± 1.4 [spread across the 1990's] | Johns et al. (2002) |
| | | | -4.8± 0.32 [Oct '20, Jul '21, Sep '21, Mar '22] | Gradone et al. (2023) |
| Antigua Passage | -2.11 | -4.06 | -3.1 ± 1.5 [spread across the 1990's] | Johns et al. (2002) |
| Guadeloupe Passage | -0.74 | -0.10 | -1.1 ± 1.1 [spread across the 1990's] | Johns et al. (2002) |
| Dominica Passage | -2.79 | -3.02 | -1.6 ± 1.2 [spread across the 1990's] | Johns et al. (2002) |
| St. Lucia Passage | -1.97 | -4.92 | -1.5 ± 2.4 [spread across the 1990's] | Johns et al. (2002) |
| St. Lucia - Trinidad | -9.51 | -16.62 | -8.6 [spread across the 1990's] | Johns et al. (2002) |
| Yucatan Passage | +20.63 | +27.08 | 23.8 [Sept '99 - Jun '00] | Sheinbaum et al. (2002) |
| | | | +23.06 [Aug '99 - Jun '01] | Candela et al. (2003) |
| | | | +27.6 [Sept '12 - Aug '16] | Candela et al. (2019) |

Updated version of Table 4 to include the Yucatan Passage transports including the most recent estimate by Candela et al. (2019) .

- Candela, J., Tanahara, S., Crepon, M., Barnier, B., and Sheinbaum, J.: Yucatan Channel flow: Observations versus CLIPPER ATL6 and MERCATOR PAM models, Journal of Geophysical Research: Oceans, 108, 2003.
- Candela, J., Ochoa, J., Sheinbaum, J., Lopez, M., Perez-Brunius, P., Tenreiro, M., Pallàs-Sanz, E., Athié, G., and Arriaza-Oliveros, L.: The flow through the Gulf of Mexico, Journal of Physical Oceanography, 49, 1381–1401, 2019
* * *
**2. Monthly climatologies for specific subregions: It could be interesting to compare monthly climatological patterns of salinity and temperature in specific subregions (either surface fields or vertical profiles). You calculated a monthly climatology in Figure 13, but that is an average for the entire interest region, which I think is not the right approach. Averaging over this large area could mask subregional biases, and you are also not discriminating important spatial variability that can be worth to describe for the region.**

Thank you for the suggestion. We have created the monthly climatology for different sub-regions (Figures included below). We have also included a revised map showing these new subregions for reference. The figures show overall similar biases in temperature across the different sub-regions. In the case of salinity, the biases are slightly larger in the eastern subregion inside the Caribbean Sea than the western subregion, similar to the biases

shown for the surface in Figures 3 and 4 of the manuscript. The meridional biases of salinity within the Caribbean Sea are similar, which also agrees with the surface biases in salinity shown in Figures 3 and 4 of the manuscript. Outside the Caribbean Sea, along the eastern side of the minor Antilles, the near-surface biases in salinity are somewhat larger during June, but otherwise similar to what we see in the surface maps. We have included in the revised manuscript the figures for the eastern CS (CS.E) and the Minor Antilles east subregions as they show biases within the Caribbean Sea are smaller than east of the Minor Antilles. The description in Section 3.3 has been extended to include a discussion regarding these two new figures.

[Figure]

Subregions for the climatologies added to the analysis and shown in figures below. This figure will be added to the Appendix.

[Figure]

Revised version of Figure 13 in the manuscript for the validation region.

[Figure]

Monthly climatology anomaly of salinity and temperature in the top 300m of the water column. The average is over the Caribbean Sea box identified as CS1 in the map above. This figure will be included in the manuscript.

[Figure]

Monthly climatology anomaly of salinity and temperature in the top 300m of the water column. The average is over the eastern side of the minor Antilles as shown in the map above. This figure will be included in the manuscript.

[Figure]

Monthly climatology anomaly of salinity and temperature in the top 300m of the water column. The average is over the northern half of box CS1 in the map above. This figure will be included in the Appendix.

[Figure]

Monthly climatology anomaly of salinity and temperature in the top 300m of the water column. The average is over the southern half of box CS1 in the map above. This figure will be included in the Appendix.

[Figure]

Monthly climatology anomaly of salinity and temperature in the top 300m of the water column: horizontal average in the eastern caribbean. This figure will be included in the Appendix.

[Figure]

Monthly climatology anomaly of salinity and temperature in the top 300m of the water column. The average is over the western half of box CS1 in the map above. This figure will be included in the Appendix.
* * *
**3. Interannual variability. Figures 14 and 15g-I revealed that the interannual transport variability is not well reproduced by the model (I disagree with the statement in lines 365-366 "While the mean inter-annual flows are well represented in CARIB12"). Does the Caribbean Current have a chaotic behavior? This should be further discussed. In addition, since the ability of the model to reproduce interannual variability is critical for the analysis of historical patterns, I wonder to what degree the model was able to simulate realistic interannual variability in temperature and salinity. Is it possible that you generate monthly time series of these variables for specific subregions and compare with observations or GLORYS?**

We agree with the reviewer's comment regarding interannual transport variability not being well reproduced by the model in some regions. We have removed the sentence in lines

365-367 of the original manuscript: "While the mean inter-annual flows are well represented in CARIB12, the model does not capture the same amplitude of variability that GLORYS12 suggests exists in some of the passages (Figures 15g-I)."

Despite this, internannual temperature and salinity variability is well represented in the model. Figure A7 shows monthly temperature and salinity anomalies in the top 300 meters of the water column. For both fields, the model captures the overall patterns of interannual variability. For example, Figure A7a and A7b show positive salinity anomalies across the last ~5 years of the simulation that are preceded by a 2-3 year negative salinity anomaly. These anomalies correspond well to similar signals in GLORYS12 (shown in Figure A7c). Similar patterns are shown for temperature with biases generally below 10% difference across the top 100 m. We revised Figure A7 (included below) and moved it to the main manuscript instead of the Appendix. We have also generated new figures for the Caribbean Sea (CS1 region in the map included in the answer to the previous comment) and the region east of the minor Antilles. The figures for the subregions show that the temperature and salinity variability is well reproduced across the subregions with differences noted in the magnitude of some of the larger anomalies.

[Figure]

New version of Figure A7. This Figure will be moved from the Appendix to the manuscript.

[Figure]

Interannual salinity and temperature anomalies in the top 300 m for region CS1 (map shown in answer to previous comment by the reviewer).

[Figure]

Interannual salinity and temperature anomalies in the top 300 m for region Minor Antilles east (map shown in answer to previous comment by the reviewer).
* * *
**4. Figure quality: The quality of the figure must be improved. You are not using any map projection to display the spatial patterns, and I think you should. I would also consider including some discretization in the colorbar to better discriminate spatial features. You could also evaluate merging several figures, like 3 and 4, 6 and 7, and 9 and 10. That may help to compare better the winter to summer changes in SSS, EKE, and MLD.**

We thank the reviewer for the feedback and agree that the surface maps in particular could be improved and in certain cases some figures could be merged. We have modified our surface field validation figures to include map projections and coastlines. We are also now using discrete colormaps instead of continuous colormaps. We also merged the following pairs of figures into single figures: figures 3 and 4, 6 and 7, and 9 and 10. Furthermore, we extended our validation of the surface current speeds to a seasonal comparison rather than

the full time mean with speed vectors shown as arrows as suggested by the reviewer in a different comment. New versions of the figures from SST and Speed are shown below. We also included a dataset for the validation of the surface currents as suggested by another reviewer.

[Figure]

New version of Figure 2 in the original manuscript with updates based on feedback by both reviewers.

[Figure]

New version of Figure 5 including feedback by both reviewers. This updated figure also shows an example of how we merged figures 3 and 4, 6 and 7, 9 and 10.
* * *
**5. Velocity patterns: In addition to the mean speed, I suggest you compare the mean velocity fields (u,v). That would add further insights about what circulation biases the model has. For example, see Figure 4 in Liu et al. (2015; http://dx.doi.org/10.1016/j.jmarsys.2015.01.007).**

Thanks for the suggestion. As shown above, we have included the velocity vectors on top of the colored speed magnitude. We believe that this provides further information on the surface currents and the model performance and the figure will be included as shown in the revision of the manuscript.
* * *
**6. Section 3.15. I am not sure if this section is worth to include in the manuscript. This is not model validation and Figure 16 is little bit hard to interpret. Unless you have actual observations to compare the simulated trajectories, I would remove this section.**

Thanks for your comment. We agree this section feels somewhat disjointed from the rest of the validation in the manuscript, hence we moved it to the Appendix, together with the description of the method used to generate the relevant figure. We also added text to the main manuscript to clarify the importance to test CARIB12's ability to reproduce known pathways of plume water intrusions into the Virgin Islands basin. While the comparison is not direct (like the validation of the surface fields for example), our results show that indeed CARIB12 is capable of reproducing these pathways. This is important because these pathways are a result of the interactions between the density front of the plume and the topography as the waters approach the eastern Caribbean Sea. Thus, reproducing the pathways indicates that CESM-MOM6 is able to reproduce these processes, with important improvements over coarse resolution climate simulations. In place of this section, we included further analysis of subregions within the Caribbean Sea.
* * *
**7. MOM6-NWA12: I wonder what difference in terms of configuration (beyond the model domain extension) has CARIB12 respect to the MOM6-NWA12 configured by Ross et al. (2024). Maybe, it could be worth to mention something about it.**

Thanks for the suggestion. We have added the following lines in Section 2.1 below Line 110 of the original manuscript: "The CARIB12 configuration has several key differences with other recent MOM6 configurations that cover a similar region, like the NWA12 configuration of Ross et al. (2023). For instance, the coupling infrastructure used in CESM-MOM6 is the Community Mediator for Earth Prediction Systems (CMEPS), whereas NWA12 uses the Flexible Modeling System (FMS) coupler. In terms of the physical configuration, a few distinctions are worth highlighting. The vertical mixing parameterizations are entirely different. As CARIB12 is a CESM-MOM6 configuration we use KPP for the boundary layer mixing parameterization which is the same scheme used in CESM. On the other hand, NA12 uses the energetics based Planetary Boundary Layer (ePBL) scheme of Reichl et al. (2018). Vertical mixing in CARIB12 is specified via the CvMix library which parametrizes vertical mixing in the interior using schemes that are different to those in NA12. For example, shear-driven mixing in CARIB12 is handled by the parameterization of Large et al. (1994), whereas NA12 applies the Jackson et al. (2008) scheme."

- Reichl, B. G., & Hallberg, R. (2018). A simplified energetics based planetary boundary layer (ePBL) approach for ocean climate simulations. Ocean Modelling, 132, 112-129.
- Large, W. G., McWilliams, J. C., & Doney, S. C. (1994). Oceanic vertical mixing: A review and a model with a nonlocal boundary layer parameterization. Reviews of geophysics, 32(4), 363-403.
- Jackson, L., Hallberg, R., & Legg, S. (2008). A parameterization of shear-driven turbulence for ocean climate models. Journal of Physical Oceanography, 38(5), 1033-1053.

---

## Author Response (AR2)

The authors would like to thank the reviewer for their feedback and comments on our manuscript. We have addressed all the reviewer comments. Below is our response detailing all relevant changes. The reviewer comments are presented in bold text followed by our response. The line number references included correspond to the revised PDF.

**An aspect that needs further clarification is the transport bias. In your answer to my comment 3, you mentioned:**
**"We have removed the sentence in lines 365-367 of the original manuscript: "While the mean inter-annual flows are well represented in CARIB12, the model does not capture the same amplitude of variability that GLORYS12 suggests exists in some of the passages." But the sentence "While the mean inter-annual flows are well represented in CARIB12" is still in the revised version. I think this statement is not well supported. The mean annual flows in Table 4 clearly show significant differences between CARIB12 and GLORYS, and the temporal variability is not clearly reproduced. I think you should at least recognize the discrepancies in the mean transport and provide an explanation about the temporal disagreement between the time series of transport.**

Thank you for the comments. While differences between CARIB12 and GLORYS are seen both for the time mean values and interannual anomalies, CARIB12 represents observed mean flows overall better than GLORYS (Table 4). Also, we cannot assess how the transports' variability in GLORYS12 (Figures 15 and 16) compares to the real ocean, as observations are limited.

In the revised manuscript, we clarify statements that are made regarding the comparison with observations versus GLORYS (for example, lines 407-410). Also, we provide a likely reason for the disagreement between CARIB12 and GLORYS (Lines 411-413).

Line 408-409 "While the mean inter-annual flows are well represented in CARIB12…" has been rewritten as follows: "While the mean flows are well represented in CARIB12 compared to observations…"

Additional relevant changes are detailed below in response to other comments by the reviewer.

**Another aspect is related to my comment 2, about monthly climatology for specific subregions. It is unclear to me what the motivation for defining the six subregions in Figure A5 are. Five of them are in the southeastern part of the Caribbean Sea, while none of them are in the northern and southwestern Caribbean Sea. Could you clarify please?**

Thanks for your question. The main region of interest in developing this configuration is the region shown in the validation figures (line 192 and Figure 2, for example). The subregion named CS1 covers an important and large area of the region of interest. The sub-regions embedded in CS1 represent distinct geographical regions with particular oceanographic features and processes.

While not shown, we also validated other subregions within the domain shown in Figure A5 and outside of it. We have not included these figures in the manuscript as they show similar patterns and biases to other regions discussed. We include in the following example figures of the validation done for the Colombian Basin and the region around the Central American rise, along with a map showing these two regions.

[Figure]

Map showing additional sub-regions in the west/southwestern CS. Example figures for these subregions (Central American rise and Colombian basin) are included below. This figure is not included in the revised manuscript and is included here as a reference for the following two figures.

[Figure]

Validation of the salinity and temperature seasonal climatology within the Central American rise subregion. This figure is not included in the revised manuscript.

[Figure]

Validation of the salinity and temperature seasonal climatology within the Colombian Basin subregion. This figure is not included in the revised manuscript.

**Specific comments are listed below.**
**15-17:**
**"We show that mean ocean mass transports across the multiple passages in the eastern Caribbean Sea compare favorably to observation-based estimates, but the model exhibits smaller variability compared to ocean reanalysis transport estimates"**
**CARIB12 underestimates the transport at Yucatan channel when compared to GLORYS and the longest observational record (Candela et al., 2019). Also, the overall comparison between ocean transport at the different passes does not show "minor" disagreements. That should be considered in the abstract. Maybe you could add "but the model exhibits smaller variability and underestimates total Yucatan channel transport when compared to ocean reanalysis estimates"**

Thank you for your comment. Following the reviewer's suggestion we have added the following text to lines 15-16:

"...but the model exhibits smaller variability and underestimates the mean Yucatan channel transport when compared to observations and ocean reanalysis estimates."

**17-19: You could be more concise about this CARBI12 vs CESM-1º comparison. Those are not surprising results.**

Thanks for the comment. We have revised the text as follows (lines 16-19):

"Furthermore, a brief comparison against a 1º CESM global ocean configuration shows that the higher resolution regional model better represents the extent and seasonality of the Amazon river plume, hence better represents near surface salinity and mixed layer depth in the CS."

**222-223: No need of a new paragraph here**

We thank the reviewer for this suggestion. We have merged the paragraphs in the revised manuscript.

**249-250: No need of a new paragraph here**

We thank the reviewer for their suggestion. We have merged these two paragraphs in the revised manuscript.

**250: The smallest biases => The smallest SSS biases**

Thank you for the suggestion. We have added "SSS" in line 228 as suggested.

**267-268: No need of a new paragraph here**

Thank you for your suggestion. We feel splitting the paragraph here makes the reading easier. The first paragraph characterizes overall biases in surface speeds across the validation region, and the second paragraph characterizes particular features in the biases of the surface flows. The resulting paragraph following the suggestion would be long and splitting further down the paragraph would interrupt the flow of the description provided there. We have decided to keep the paragraphs here as they are.

**273: "represents well the inflow" => "represents well the surface inflow"**

Thank you for your suggestion. We have edited line 252 to reflect this suggestion:

**345: "Figure 4 shows" => "Figure 8 shows"**

Thank you for pointing out this error, we have corrected this reference in line 315.

**372: Figure A5: The borders of these overlapping regions are somewhat unclear. Maybe, you could depict with a distinct color each specific regional border. Also, you may want to provide a justification about why you are defining those regions. The selection of regions seems to me somewhat arbitrary, and not fully informative of the main interest region.**

Thank you for the comment. The selection of the sub-regions presented in the manuscript and Appendix is representative of the main interest region for our work, indicated e.g. at line 192. We have reviewed Figure A5 following this and other comments by the reviewer. We also show examples of validations of additional sub-regions in a previous comment by the reviewer.

[Figure]

Updated version of Figure A5 in the revised manuscript.

**371-378: No mention to Figure 12. Also, I am not convinced that you need to include three figures (Figures 10, 11, and 12) in the paper main body (which show very similar patterns) to describe the vertical variability of temperature and salinity at the seasonal timescale.**

Thank you for the comment. The omission of Figure 12 was a grammatical error. We agree with the reviewer that the sub-regions presented (and other subregions not presented) show very similar patterns. We now moved Figures 11 and 12 to the Appendix, but we are keeping the text at Lines 342-349 describing reduced biases within the CS as that is relevant for the main region of interest in developing this configuration.

**433-450: There is no mention to the evident weak correlation between CARIB12- and GLORYS-derived patterns at the intraseasonal and interannual timescales (Figures 15 and 16g-l). Something about this temporal mismatch should be added.**

Thanks for the comment. Mismatches between CARIB12 and GLORYS12 mean transports and variability are described in section 3.4 (e.g. lines 368-369, 371, 380-389, 390-398, 399-400, and 407-409). In addition, in lines 409-415 we address potential differences in the models leading to these discrepancies. We also clarify that we cannot assess how real the variability shown in GLORYS12 is as the observational record for the passages in the eastern CS is short. We have rewritten lines 407-410 as follows:

"While the time mean flows are well represented in CARIB12 compared to observations, the model does not capture the same amplitude and frequency in flow variability that GLORYS12 suggests exists in some of the passages at sub-seasonal and inter-annual time scales (Figures 15 and 16g-l)."

And in line 413:
"Continuous observations would be needed to better assess how CARIB12 and GLORYS12 represent variability across timescales…"

**442: "While the mean inter-annual flows are well represented in CARIB12". This statement is not supported by Table 4. The differences between the CARIB12- and GLORYS-derived transport are not minor.**

Thanks for your comment, We have addressed this in the reply to a previous comment.

**481-482: You could insert this statement in section 3.1, maybe including an additional Figure in the Supplement. Otherwise, remove it.**

Thanks for the comment. We have moved the statement to lines 283-284 under Section 3.1.2. While we do not include a figure regarding validation of tides given several limitations (resolution, topography, lack of filtering for additional contributors to water elevations), we believe the text in question could be useful to the reader and users of this and other configurations of CESM-MOM6.

**482-483: As mentioned before, I disagree with this statement. You should recognize some discrepancies.**

Thanks for the comment. We reworded the text in lines 425-426 as follows:

"The mean flows are also well represented compared to observations, but CARIB12 shows lesser variability when compared to GLORYS12 flows and underestimates the mean Yucatan channel transport."

**492-497: You should tone down this paragraph. There are important differences between the transport series from CARIB12 and GLORYS, in terms of mean variance as well as temporal correlation.**

Thank you for the comment. The lines referenced describe what the model does well and doesn't do well in terms of the transports when compared to observations and GLORYS12. We have reworded the following lines (438-440) to address the reviewer's comment:

"The seasonal transports in CARIB12 compare overall well with the GLORYS12 reanalysis, yet GLORYS12 exhibits larger variability at sub-seasonal and interannual timescales."

**514-517: You are attributing the biases mainly to two sources, but this is no supported by any analysis. If this is only speculation, I would use "might" instead of "can".**

Thanks for the suggestion, we rewritten the sentence in line 452 as follows: "The main biases in CARIB12 may be attributed to two sources."